# Dataset of 1-km cropland cover from 1690 to 1999 in Scandinavia

Xueqiong Wei[1], Mats Widgren[2], Beibei Li[3], Yu Ye[4,5], Xiuqi Fang[4,5], Chengpeng Zhang[5], Tiexi Chen[1]

[1] School of Geographical Sciences, Nanjing University of Information Science and Technology, Nanjing 210044, China

[2] Department of Human Geography, Stockholm University, SE-106 91 Stockholm, Sweden

[3] Division of Science and Technology, Nanjing University of Information Science and Technology, Nanjing 210044, China

[4] Key Laboratory of Environmental Change and Natural Disaster, Ministry of Education, Beijing Normal University, Beijing 100875, China

[5] Faculty of Geographical Science, Beijing Normal University, Beijing 100875, China

*Correspondence to*: Xueqiong Wei (xueqiong.wei@nuist.edu.cn)

## Abstract

Spatially explicit historical land cover datasets are essential not only for simulations of climate and environmental dynamics but also for projections of future land use, food security, climate, and biodiversity. However, widely used global datasets are developed for continental-to-global scale analysis and simulations and their accuracy depends on the verification of more regional reconstruction results. This study collected cropland area data of each administrative unit (parish/municipality/county) in Scandinavia from multiple sources. The cropland area data were validated, calibrated, interpolated, and allocated into 1 km × 1 km grid cells. Then, we developed a dataset with spatially explicit cropland area from 1690 to 1999. Results indicated that the cropland area increased from 1.82 million ha to 6.71 million ha from 1690 to 1950, then decreased to 5.90 million ha in 1999. Before 1810, cropland cover expanded in southern Scandinavia and remained stable in northern Scandinavia. From 1810 to 1910, northern Scandinavia experienced slight cropland expansion, and the cropland area increased rapidly in the southern part of the study area before changing slightly. After 1950, the cropland areas began to decrease in most regions, especially in eastern Scandinavia. When

comparing global datasets with this study, although the total Scandinavia cropland area is in agreement among HYDE 3.2, PJ, and this study, the spatial patterns show considerable differences, except for in Denmark between HYDE 3.2 and this study. The dataset can be downloaded from https://doi.org/10.1594/PANGAEA.926591 (Wei et al., 2021).

## 1 Introduction

The Anthropocene has been defined as a new epoch of geologic time, partly because human-influenced land is a major component of anthropogenic global changes in the Earth System (Crutzen and Stoermer, 2000; Lewis and Maslin, 2015; Verburg et al., 2016). During AD 800–AD 1700, according to Pongratz et al. (2008), 5% of the area covered by natural vegetation was under human land use, compared to 44% in the following 300 years. Anthropogenic land cover change (ALCC) may have caused feedbacks to the climate system through modifying the surface roughness, surface albedo, latent heat flux, and river runoff, and through changing atmospheric $CO_2$ concentration (Foley et al., 2005; Pongratz et al., 2009a, 2010; Houghton et al., 2012, 2018; Ciais et al., 2013; Myhre et al., 2013; Yang et al., 2015; Liu et al., 2016; Kaplan et al., 2017; Le Quéré et al., 2018). The conclusions of climate and environmental dynamics were all made by using spatially explicit historical land cover datasets. Historical land cover change information is also essential as a baseline analysis for projections of future land use, food security, climate, and biodiversity (Foley et al., 2011; Hurtt et al., 2011; Ellis et al., 2012; Brovkin et al., 2013; Fuchs et al., 2015; Mehrabi et al., 2018).

To produce spatially explicit land cover datasets covering long periods, researchers used multiple sources, such as satellite data, historical statistics, historical maps, and pollen records. Based on combined sources and hindcasting methods, the Center for Sustainability and the Global Environment (SAGE) (Ramankutty and Foley, 1999, 2010) and the History Database of the Global Environment (HYDE) (Klein Goldewijk, 2001) were produced as two representative datasets of global land use/cover. Based on SAGE and historical population data, the PJ dataset covers AD 800–AD 1700 (Pongratz et al., 2008). Subsequently, ALCC from 8000 BP to AD 1850 (KK10) was reconstructed (Kaplan et al., 2009, 2011). These global datasets were widely used in simulations of global and regional climate change or

carbon budget because of their spatial resolutions and long-term coverages (Foley et al., 2005; Olofsson and Hickler, 2007; Strassmann et al., 2008; Pongratz et al., 2009b, 2010; Van Minnen et al., 2009; Arora and Boer, 2010; Hurtt et al., 2011; Kaplan et al., 2011; Brovkin et al., 2013; Yan et al., 2017; Zhang et al., 2017; Le Quéré et al., 2018).

However, using population data to estimate historical land use may induce large uncertainties and limitations in presenting regional scale details (Klein Goldewijk and Verburg, 2013). Many regional land use reconstructions illustrated that global datasets had non-negligible discrepancies in reflecting regional spatial land use patterns historically, especially for cropland. Historical document-based reconstructions concluded that SAGE, HYDE, and PJ

had drawbacks in capturing the spatial distribution of historical cropland change in China (Li et al., 2010; Zhang et al., 2013; Li et al., 2016, 2019; Wei et al., 2019). In the US, the HYDE maps substantially underestimate crop density in high cropland coverage regions but overestimate it in the low-density areas for 1850–2016 (Yu and Lu, 2018). Neither KK10 nor HYDE captures the fine-scale spatial pattern of open land as inferred from the pollen-based

land cover reconstructions in Europe for four preindustrial time points (Kaplan et al., 2017). Uncertainties in global datasets could translate into higher uncertainties in quantifying the climate and environmental effects of ALCC at both local and regional scales (Yang et al., 2018; Lejeune et al., 2018; Yu et al., 2019). Therefore, the PAGES LandCover6k and related projects aim to improve ALCC history at both regional and global scales based on empirical

data (Gaillard et al., 2015a; Widgren, 2018a). Errors can be assessed or corrected using the regional quantitative reconstructed land cover data and regional agrarian history maps (Widgren, 2018b; Fang et al., 2020). However, due to the lack of scrutinized and published datasets at high spatial resolutions, it is impossible to reconstruct historical regional cropland change in all parts of the world.

Scandinavia (Sweden, Norway, and Denmark) has good historical cropland area data at the parish/municipality/county level. Scandinavia is located at a high latitude, and the cold climate is relatively unfavorable for crop growth. The soil suitable for cultivation is mainly distributed on the plains, around river valleys and lakes. The pollen-based reconstructions indicated that the cereal crop cover percentages were very high in southern Sweden and

Denmark (Nielsen et al., 2012; Gaillard et al., 2015b). Large forests and small agricultural land areas distinguish Scandinavia from continental Europe (Anderberg, 1991). Though arable land accounts for a small proportion of the land area, farmers are historically drivers of economic and social change (Gadd, 2011; Jansson, 2011). Over the past few centuries, agriculture in Scandinavia has undergone significant changes (Anderberg, 1991; Almås, 2004; Vejre and Brandt, 2004; Jansson, 2011). During the middle of the seventeenth century, peasants comprised most of the population, and farms were scattered and close to the village (Almås, 2004; Cui et al., 2014). The most substantial population growth has occurred since the beginning of the nineteenth century. Because of increasing food consumption, technology introduction, and land reforms, agricultural land use accelerated. The agrarian revolution altered the agricultural landscape as farms moved out of the villages and were consolidated to form single blocks. With the agricultural revolution, more land was cultivated and arable land area expanded at the expense of meadows and through drainage projects (Jansson, 2011). Heathland was also cultivated and afforested. Sweden transformed from a grain-importing country to considering grain as one of the most important export commodities (Olsson and Svensson, 2010). In response to the intensification and specialization of agriculture and forestry during the twentieth century, land use development changed gradually. An increase in forest areas and a decrease in arable land was observed.

Studies of the agricultural history in Scandinavia have mainly concentrated on agricultural policy, agricultural economy, settlement and population, and landscape history (Anderberg, 1991; Olsson et al., 2000; Almås, 2004; Vejre and Brandt, 2004; Gustavsson et al., 2007; Olsson and Svensson, 2010; Jansson, 2011; Eriksson and Cousins, 2014; Levin et al., 2014; Mazier et al., 2015; Jacks, 2019). However, spatially explicit historical land cover data for Scandinavia are scarce. Pollen-based quantitative reconstruction of Holocene regional vegetation cover in Europe presented a $1° \times 1°$ spatial scale dataset for climate modeling (Trondman et al., 2015). The agricultural land data for five Holocene time points are available. To facilitate simulation studies with high-precision regional input data, Li et al. (2013) developed the cropland cover change dataset from 1875 to 1999 in Sweden and Norway. Materials from mainly two sources were collected, including Swedish data from the official

agricultural statistics and the data provided by Norwegian Social Science Data Services (NSD). Methods of seed-cropland conversion, data interpolation and allocation, and data gridding were used to produce the cropland dataset at the spatial resolution of 0.5 degrees. At present, the highest resolution of the existing historical cropland cover dataset for Scandinavia

is 5′ × 5′ from HYDE 3.2 (Klein Goldewijk et al., 2017).

This study's main objective is to provide a longer historical gridded cropland dataset for 1690–1999 with high precision in Scandinavia, by reconstructing cropland area at the parish, municipality, and county levels and allocating cropland area into 1 km ×1 km grid cells. Compared with existing datasets, our newly developed dataset of cropland cover will provide

more detailed information on spatial patterns of historical cropland change in Scandinavia.

## 2 Data sources

In this study, cropland from multiple data sources were first gathered for further allocation. Meanwhile, administrative division maps and digital elevation model (DEM) data were also used. Land cover in 2000 based on remote sensing methods was selected to constrain

historical cropland allocation. More details are as follows.

### 2.1 Cropland data

Besides the Scandinavian Peninsula (Sweden and Norway) cropland data after 1875 provided by Li et al. (2013), the cropland data used in this study were from different sources. Data sources in this study are shown in Table 1.

For Sweden, all the data before 1875 were from the *Svensk Nationell Datatjänst* (Swedish National Data Service, termed SND, https://snd.gu.se/en/catalogue/study/SND0910). Based on tax records, historical maps, land survey records, and famer inventories, Palm et al. (2014) developed an agricultural database (The database Sweden 1570–1810: population, agriculture, land ownership), which covers all parishes within Sweden's contemporary boundaries and the

periods between 1570 and 1810. In SND's database, cropland was called "åker" in Swedish, which was also used in the data sources from 1875 to 1999 in Sweden. "Åker" from SND included land under temporary crops, land under temporary meadows and pastures, and land

temporarily fallow. However, the unit of "åker" in the dataset referred to barrels of seeds and not hectares.

For Norway, cropland data in 1665 and 1723 were from *Statistiske studier over folkemængde og jordbrug i Norges landdistrikter i det syttende og attende aarhundrede* (Statistical studies on population and agriculture: in the rural areas of Norway in the seventeenth and eighteenth centuries, Aschehoug, 1890). Cropland data in 1809 was from Hovland's study (1978). In the above two sources, the volumes of different seed types, such as wheat, rye, barley, oat, peas, and potatoes were recorded (land under temporary crops). The volumes of seeds in each county (*amt*) were presented.

For Denmark, cropland data were from multiple sources. Data in 1688 were based on the *Atlas over Denmark: Historisk-Geografisk Atlas* (Atlas over Denmark: Historical-Geographic Atlas, Dam and Jakobsen, 2008). Dam and Jakobsen's map showed a cropland fraction of each "*ejerlav*" (area under a village, a manor, or a group of single farms) in Denmark. The cropland was called "*ager*" in Danish, referring to the total area under crop rotation, including the lands under temporary crops, temporary meadows and pastures, and temporarily fallow land. Pia Frederiksen from Aarhus University provided the cropland data for 1800, 1881, and 1998, based on data developed by Jørgen Rydén Rømer, Aalborg, Bernd Münier, and Morten Stenak, Roskilde (Odgaard and Rømer, 2009). Data for 1800 were from the map Videnskabernes selskabs kort (VSK) in 1762–1806 and further developed from agricultural statistics. The data for 1881 and 1998 were merged from maps and national statistics. They had the smallest spatial unit of "*sogn*" (parish). The cropland was called "*agerjord*" and it could be divided into two subgroups, namely "*Besået areal*" (the lands annually sown with various one-year or two-year crops) and "*Græs i omdrift*" (the lands for temporary meadow or fallow before the land was again plowed for sowing). "*Besået areal*" and "*Græs i omdrift*" correspond to temporary cropland, and the lands with temporary meadows and temporarily fallow in 1688, respectively. The cropland area data in 1907 were from *Statistisk Aarbog 1912* (Statistical Yearbook 1912, Danmarks Statistik, 1912) and its spatial resolution was the "*amt*" (county). The cropland area data in 1936, 1950, and 1980 were from agricultural statistics of *Statistiske Meddelelser* (Danmarks Statistik, 1936, 1950, and 1980). The spatial

resolutions of cropland data in 1936, 1950, and 1980 were the "*amt*" (county), "*amtsrådskreds*" (county council), and "*kommuner*" (municipality), respectively. The areas under temporary crops, under temporary meadows, and temporarily fallows were listed in tables by Danmarks Statistik.

5          Table 1 Description of cropland data sources used in this study

| Data sources | Spatial coverage | Years | Spatial Resolution | Categories included in recorded cropland | Reference |
|---|---|---|---|---|---|
| *Sockenvis jordbruksstatistik* | Sweden | 1690, 1750, 1810 | Parish | A, B, C (*Åker*) | SND |
| *Statistiske studier over folkemængde og jordbrug i Norges* | Norway | 1665, 1723 | County | A | Aschehoug, 1890 |
| *Historisk Tidsskrift* | Norway | 1809 | County | A | Hovland, 1978 |
| *Atlas over Denmark: Historisk-Geografisk Atlas* | Denmark | 1688 | *Ejerlav* | A, B, C (*Ager*) | Dam and Jakobsen, 2008 |
| *Danske landbrugs-landskaber gennem 2000 år* | Denmark | 1800, 1881, 1998 | Parish | A, B, C (*Agerjord*) | Odgaard and Rømer, 2009 |
| *Statistisk Aarbog 1912* | Denmark | 1907 | County | A, B, C, D | Danmarks Statistik, 1912 |
| *Statistiske Meddelelser 1936, 1950 and 1980* | Denmark | 1936, 1950, 1980 | County; County council; Municipality | A, B, C, D | Danmarks Statistik, 1936, 1950 and 1980 |
| *Cropland in Scandinavian Peninsula* | Sweden, Norway | 1875, 1910, 1930, 1950, 1980, 1999 | County/0.5-degree | Sweden: A, B, C (*Åker*) <br><br> Norway: A, B, C, D, E | Li et al., 2013 |

Notes: A—Areas under temporary crops; B—Areas under temporary meadows and pastures; C—Land temporarily fallow; D—Areas under permanent crops; E—Permanent grassland and meadow

## 2.2 Administrative division maps

The digital version of the administrative division maps of Sweden was initially developed at the National Archives of Sweden (*Riksarkivet*). It was later revised at the Department of Human Geography, Stockholm University. Ulf Jansson kindly provided the version used here. The administrative division maps of Norway were from Norwegian Centre for Research Data (termed NSD, https://nsd.no/nsd/english/). The base maps of Denmark were downloaded from

the HisKIS network (http://hiskis2.dk/?page_id=110) and Danish Geodata Agency (http://download.kortforsyningen.dk/).

## 2.3 Satellite-based data

Satellite-based land cover data 2000 were selected as a reference to constrain the spatial distribution of historical cropland allocation. There are several well developed and widely

used land cover datasets (Zhang et al., 2019). The satellite-based land cover data in 2000 used in this study were provided by the CORINE Land Cover (CLC2000, https://land.copernicus.eu/pan-european/corine-land-cover/) (Büttner, 2014), which has been well evaluated in Europe (Feranec et al., 2010). Although CLC1990 data are also provided, only CLC2000 contains the earliest data covering our study region in the CLC data series. It

consists of an inventory of land cover in 5 main categories and 44 classes. The details of CLC2000 application in our model are explained in Section 3.2.

The DEM used in this study was from the NASA Shuttle Radar Topographic Mission (SRTM, https://cgiarcsi.community/data/srtm-90m-digital-elevation-database-v4-1/) 90 m DEMs.

## 3 Methods

There are two main steps to develop the 1 km spatial resolution gridded cropland dataset for 1690 to 1999. 1) Original multiple-source cropland area data were collected to reconstruct the cropland area of each administrative unit. 2) Using an allocation model that featured the use of remote sensing land cover data as a constraint, cropland in administrative units were

further allocated into 1 km × 1 km grid cells. The flow chart of the methods is described in Figure 1.

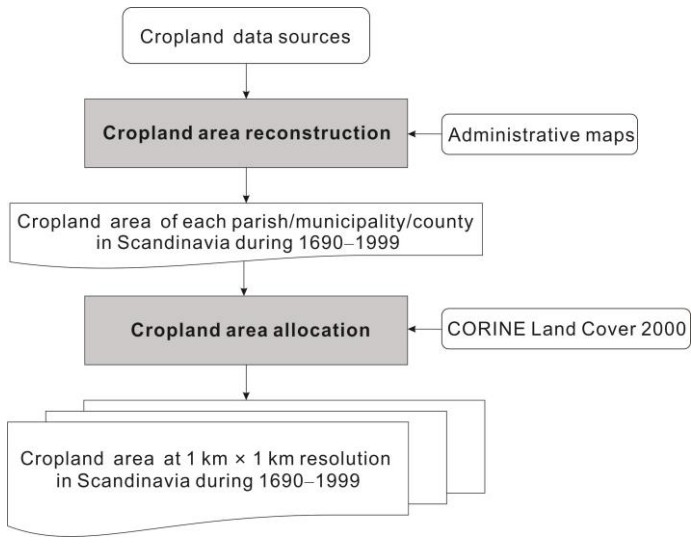

Figure 1 Flow chart of the methodology

## 3.1 Cropland area reconstruction at the parish/municipality/county levels

Based on the multiple sources, we collected cropland area data of each parish/municipality/county in Scandinavia from 1690 to 1999. Cropland data recorded as barrels of seeds were converted to cropland area in Sweden and Norway before 1875. By comparing between statistics and census in 1999, cropland area data from statistics were calibrated. After time points selection, missing data were interpolated, and the cropland area at the parish/municipality/county levels in Scandinavia from 1690 to 1999 was developed (Figure 2).

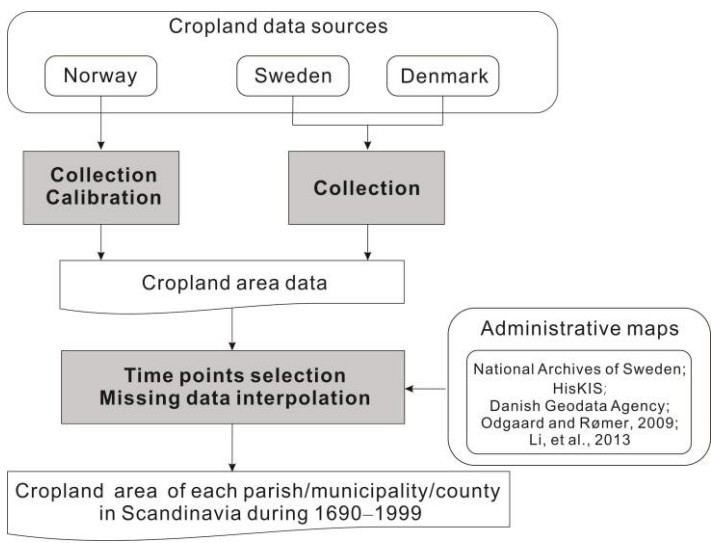

Figure 2 Flow chart of reconstructing the cropland area at the parish/municipality/county

levels

### 3.1.1 Cropland data collection and calibration

The category "cropland" defined by FAO (http://www.fao.org/) was used in this study. Thus, our cropland includes arable land (areas under temporary crops, temporary meadows and pastures, land temporarily fallow) and areas under permanent crops. Non-cropland area, such as permanent meadows, were excluded.

Because the unit of Swedish data in 1690, 1750, and 1810 from SND was barrels of seeds and not area, we used the formula of 1 barrel of seeds = 4936 $m^2$ of cropland = 0.4936 hectare (ha) of cropland (Cardarelli, 2003) to obtain the cropland area of each parish. The Norwegian data in 1665, 1723, and 1809 also provided the volume of seeds but not the cropland area. We collected the relationships between the volume of seeds and the cropland area from four sources (Table 2). The values of liter per *maal* (1 *maal* = 0.1 ha) for seven types of seeds were close to each other, except for the value from NSD. The closest year to the three time points of 1665, 1723, and 1809 was 1835. Because agricultural development reduced the use of seeds per *maal* gradually, we chose the values of liter per *maal* in 1835 from *Statistiske Oversigter 1914* (Aschehoug, 1914). For data sources in the remaining years and in Denmark, area units were used, and we unified the area units as ha.

Table 2 Relation of *maal* to liter from different sources in Norway

| Sources | Oversigt over det norske landbruks utvikling siden 1750 (Klokk, 1920) | | | Norges Landbrug i Dette Aarhundrede (Smitt, 1888) | Statistiske oversigter 1914 (Aschehoug, 1914) | | NSD |
|---|---|---|---|---|---|---|---|
| Liter per maal\Year | 1896–1900 | 1901–1905 | 1907 | 1866–1875 | 1835 | 1865 | 1907 |
| *Hvete* (wheat) | 29.0 | 27.8 | 28.2 | 26.4 | 25.0 | 25.2 | 39.2 |
| *Rug* (rye) | 20.4 | 20.8 | 22.5 | 19.5 | 19.5 | 19.3 | 31.2 |
| *Byg* (barley) | 32.8 | 31.9 | 32.2 | 33.4 | 34.8 | 34.7 | 44.8 |
| *Havre* (oats) | 46.9 | 45.5 | 45.2 | 51.4 | 52.8 | 53.2 | 62.9 |
| *Blandkorn* (mix) | 42.6 | 40.5 | 40.2 | 47.3 | 47.3 | 47.0 | 55.9 |
| *Erter* (peas) | 30.4 | 26.8 | 30.9 | 29.2 | 30.6 | 30.9 | 42.9 |

| | | | | | | | |
|---|---|---|---|---|---|---|---|
| *Poteter* (potato) | 302.2 | 306.8 | 302.7 | 290.5 | 290.5 | 290.3 | 420.7 |

Cropland data from SND were based on tax records, historical maps, land survey records, and farmer inventories. Cross-validation of data from multiple sources ensured data reliability. However, land under permanent crops were excluded in SND's data, which underestimated the total cropland area before 1875 in Sweden. However, no additional historical records were found regarding the size of land under permanent crops. Using the same coefficient in different years to estimate land area under permanent crops will increase the discrepancy of the cropland area. Moreover, based on the Farm Structure Survey (FSS, https://ec.europa.eu/) in 1999 and 2010, the total land size under permanent crops accounted for only approximately 0.1% of all cropland area in Sweden. Thus, we used the converted area from the barrels of seeds as the cropland area before 1875 in Sweden. For the collected cropland area data after 1875 in Sweden, Li et al. (2013) evaluated the statistics employed and ensured the data accuracy. Cropland area data of Denmark in 1688, 1800, and 1881 were also validated using historical maps when they were generated (Dam and Jakobsen, 2008; Odgaard and Rømer, 2009). As the census from the FSS showed that the area under permanent crops accounted for less than 1% of all cropland in Denmark, we used "ager" and "agerjord" as cropland in 1688, 1800, 1881, and 1998. Danish statistics recorded the land use areas of all crops, temporary meadows and pastures, and fallow land. Therefore, we selected the land area categorized as cropland according to the FAO and calculated its total area.

The total cropland area collected by Li et al. (2013) from statistics in Norway was $1.04 \times 10^6$ ha in 1999, which was 38% larger than the total area of arable land and land under permanent crops in 2000 from the FSS census but was the same as the total area of arable land, land under permanent crops, and land under permanent grassland and meadow from the FSS. Thus, we re-collected the arable land area and land area under permanent crops from the 1999 statistics (NSD kommunedatabase, https://kdb.nsd.no/kdbbin/kdb_start.exe) used by Li et al. (2013). We found the total cropland area provided by the FSS census in 2000 was 1.18 times larger than our collected statistics. Thus, a calibration coefficient of 1.18 was used to compensate for estimating the cropland area in Norway from official statistics from 1665 to 1999.

### 3.1.2 Time points selection and data interpolation

Based on the years when cropland area data are available in Sweden, Norway, and Denmark, we selected nine times to represent the cropland change trend from 1690 to 1999. Compared to the data sources in Norway and Denmark, the cropland data from Swedish data sources were the most abundant and complete. Thus, the nine time points were based on the data sources in Sweden, which were 1690, 1750, 1810, 1875, 1910, 1930, 1950, 1980, and 1999. The years of 1690, 1750, and 1810 correspond to 1665, 1723, and 1809 from the data sources in Norway. In Denmark, 1690, 1810, 1910, 1930, 1950, 1980, and 1999 correspond to 1688, 1800, 1881, 1912, 1936, 1950, and 1998, respectively.

To map the spatial patterns of cropland distribution in Sweden before 1875, we used the parish level administrative map from 1750. We linked the datasets during 1690–1810 to the base map in 1750 based on parish code. Four cities (*stad*) in the datasets failed to connect with the corresponding base map, and fourteen parishes on the base map also lacked cropland data. After checking their administrative codes, a relationship between the four cities (*stad*) and fourteen parishes was found. Then, all the 2390 parishes were associated with their corresponding cropland area data.

For Denmark, cropland area data in 184 (total 8083) "*ejerlavs*" were missing in 1688. We assumed that neighboring "*ejerlavs*" with similar terrain had the same cropland fraction and cropland growth rate. Then, the missing data were interpolated based on the cropland fractions of their neighboring "*ejerlavs*" in 1688 and the cropland area changes from 1688 to 1800. Using the same method, we also interpolated 56 (total 1705) missing data in 1800 and 2 missing data (total 1719) in 1881. We selected the administrative map in 1688 from the HisKIS network as the cropland data base map. Cropland area data in 1800, 1881, and 1999 also had corresponding base maps (Odgaard and Rømer, 2009). The base maps for data in 1936, 1950, and 1980 were from the *Danish Geodata Agency* (https://eng.gst.dk/). There was no cropland area record of Denmark in 1750. Therefore, we assumed that the cropland area change rate from 1690 to 1810 was constant and computed each parish's cropland area in 1750.

Because the cropland data before 1875 was at the county (*amt*) level and the administrative division did not change dramatically from 1690 to 1875 in Norway, the administrative map in 1875 from NSD's kommunedatabase was used for data during 1690–1810 as the base map. Following the above steps, the cropland area of each administrative unit at the nine time points was connected to the corresponding administrative maps.

## 3.2 Cropland area allocation into grid cells

As the cropland area data of each parish/municipality/county cannot be used as input for the climate and environment simulations directly, we developed cropland area allocation models and allocated the cropland area into 1 km × 1 km grid cells. The allocation process included two essential parts, cropland distribution factors and the maximum extent constraints. In this work, elevation and slope were selected as the distribution factors, and the maximum extent constraints were defined based on the CLC2000 data.

We analyzed the factors affecting cropland distribution. In previous studies, elevation, slope, climate, soil, water, and population were used as causes related to the change in cropland spatial distribution (Klein Goldewijk et al., 2011; Li et al., 2016; Paudel et al., 2017). People tend to prefer starting from areas with lower elevations and gentler slopes when cultivating the land. Land with high elevations and slopes has negative characteristics that constrain cropland cultivation; therefore it will only be used after low-elevation and gentle-slope land has been cultivated. In Scandinavia, there is little climate difference in each parish/municipality/county. We assumed the impact of climate on cropland distribution had been included in the effects of elevation and slope in each parish/municipality/county. Soil properties such as texture, fertility, and organic-matter content impact suitability for growing crops, but it is not a limitation for land cultivation in most areas. According to statistics, Scandinavia's population constantly increased from the 17th century to the present (SCB, SSB, and Statistics Denmark), while the cropland area decreased after 1950. Population growth was an important reason for cropland area increase before 1950, especially in Sweden and Denmark. However, the spatial distribution of population data before 1950 at a high spatial resolution is hard to obtain. Thus, elevation and slope were selected as the factors in the cropland area allocation model. Using the NASA SRTM 90 m DEMs, we resampled the

DEMs in Scandinavia to 1-km resolution. The values of elevation and slope were normalized using the following formulas:

$$E'(i)=\frac{E_{max}-E(i)}{E_{max}-E_{min}} \tag{1}$$

$$S'(i)=\frac{S_{max}-S(i)}{S_{max}-S_{min}} \tag{2}$$

where $E'(i)$ and $S'(i)$ are the normalized elevation and slope value of grid $i$, $E(i)$ and $S(i)$ are original elevation and slope value of grid $i$, $E_{max}$ and $S_{min}$ are the maximum elevation and slope value of grid $i$ in Scandinavia, and $E_{min}$ and $S_{min}$ are the minimum elevation and slope value of grid $i$ in Scandinavia.

The value of land cultivation suitability $Suit(i)$ of grid $i$ is calculated using the following
formula:

$$Suit(i)=Mcrop(i)\times E'(i)\times S'(i) \tag{3}$$

where $Mcrop(i)$ refers the maximum extent of cropland, which is a crucial component for allocation models. In this study, $Mcrop(i)$ is defined using the CLC2000 data.

The weight of grid $i$ for cropland area allocation ($w(i)$) and the cropland area of grid $i$ ($Crop(i)$)
become:

$$w(i)=\frac{Suit(i)}{\sum_{i=1}^{n}Suit(i)} \tag{4}$$

$$Crop(i)=w(i)\times Crop(p_n) \tag{5}$$

where $Crop(P_n)$ is the total cropland area of administrative unit ($P_n$). The total weight of each administrative unit for cropland area allocation is 1.

In previous studies, the maximum extents of cropland ($Mcrop$) in modern times have been used to allocate historical cropland area (Li et al., 2016; Wei et al., 2019) because usually the

area of cropland in modern times is larger than that in historical periods. However, this is a source of error in Scandinavia as many croplands were converted to forest during the 20th century. Beyond the maximum extent of modern cropland, more marginal lands may have been cultivated. Considering Scandinavia's agricultural history and urbanization development, we used different maximum extents of cropland during 1690–1950, in 1980, and 1999.

Because urban land is usually built on rural and cultivated land, urban land cover maps from CLC2000 were also used to build the maximum cropland cover extent map. In Scandinavia, although industrialization accelerated after 1870, the urbanization rate was still less than 30% before 1930. After 1960, urbanization was even faster, and its rate reached over 50% in 1990 (Clark, 2009). Thus, the urban land cover map was only used during 1690–1950. The cropland area was allocated to grid cells following the process shown in Figure 3.

(1) The maximum extent of cropland in the first step allocation is defined as the sum of "Arable land," "Permanent crops," and "Discontinuous urban fabric" from CLC2000. $Mcrop\_I(i)$ and $Mcrop\_I(P_n)$ refers to the maximum extent of cropland in grid $i$ and the administrative unit $P_n$, respectively. We compared our collected cropland area of administrative unit $P_n$ ($Crop(P_n)$) with $Mcrop\_I(P_n)$. If $Crop(P_n) <= Mcrop\_I(P_n)$, $Crop(P_n)$ was allocated into grid cells using formulas (3), (4), and (5) mentioned above, where $Mcrop\_I(i)$ is $Mcrop(i)$ in the formula (3). Otherwise, if $Crop(P_n) > Mcrop\_I(P_n)$, $Mcrop\_I(P_n)$ was allocated and the rest ($Rest1(P_n)$, calculated as $Crop(P_n) - Mcrop\_I(P_n)$ ), went into the next step (Figure 3).

In the second step, $Rest1(P_n)$ was compared with $Mcrop\_II(P_n)$. If $Rest1(P_n) <= Mcrop\_II(P_n)$, $Rest1(P_n)$ was allocated into grid cells using formulas (3), (4), and (5), where $Mcrop\_II(i)$ is $Mcrop(i)$ in formula (3). Otherwise, $Mcrop\_II(P_n)$ was allocated. The rest ($Rest2(P_n)$, calculated as $Rest1(P_n) - Mcrop\_II(P_n)$), was compared with $Mcrop\_III(P_n)$ and allocated following the same procedure as the first and second steps. $Mcrop\_IV(i)$ and $Mcrop\_V(i)$ were used in formula (3) in turn, until the cropland areas in all administrative units were allocated to grid cells.

For the cropland area in 1980, as the urbanization rate reached approximately 50%, more cropland was converted to urban land and the maximum extent of cropland in the first allocation step was defined as the sum of "Arable land" and "Permanent crops" from CLC2000. *Crop(P_n)* was allocated to grid cells using the allocation model in Figure 3. For cropland area in 1999, following the allocation steps in Figure 3, *Crop(P_n)* was allocated to *Mcrop_I(i)* (defined as all "Arable land" and "Permanent crops" of grid *i*), *Mcrop_II(i)* (defined as "Complex cultivation patterns" of grid *i*) and *Mcrop_III(i)* (defined as "Land principally occupied by agriculture, with significant areas of natural vegetation" of grid *i*) in turn.

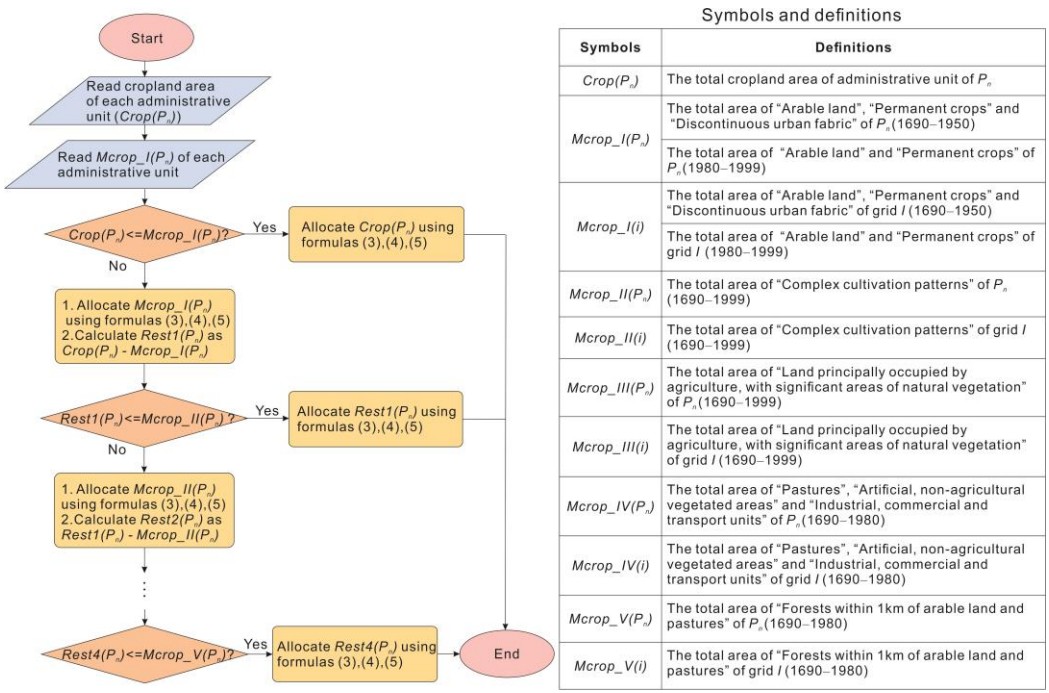

Figure 3 Flow chart of cropland area allocation model

### 3.3 Cropland data comparison

Because global datasets were produced based on national level cropland area while cropland areas at the parish/municipality/county levels were used to spatialize cropland distribution by us, cropland datasets from this study could be used for assessing the accuracy of global datasets. As the total cropland area is small in Scandinavia, the relative difference ratio (*RD*) was used to identify the differences between different datasets as follows:

$$RD = \frac{C_{globe}(t) - C(t)}{C(t)} \times 100\% \tag{6}$$

where $C_{globe}(t)$ is the cropland area from global datasets and $C(t)$ is the cropland area from this study.

## 4 Results

Because the cropland area only accounted for less than 8% of Scandinavia's total land area in 2000, small changes were difficult to observe from the 1 km × 1 km resolution cropland maps, especially in northern Scandinavia. However, we found the total cropland area from 1690 to 1999 showed a phased change. Significant changes in the spatial distribution of cropland in southern Scandinavia and northern cropland expansion were also shown clearly on our maps.

### 4.1 Changes in the total cropland area in Scandinavia

The total cropland area change in Scandinavia during 1690–1999 is shown in Figure 4. Overall, cropland area developed slowly before 1810, and kept developing rapidly or even increasing until the beginning of the 20th century, then remained stable for around 40 years before dropping in 1999. The cropland area change process could be divided into four stages: (1) Slight increase in 1690–1810, with an annual growth rate of 0.37% on average. In 1690, the cropland area was 1.82 million ha, accounting for 2.07% of the land area. The cropland area rose steadily to 2.84 million ha in 1810. (2) Rapid increase during 1810–1910, with a growth rate of 0.82% on average annually. After 1810, the cropland area rose dramatically for the next century, reaching 6.43 million ha in 1910. 3) Steady rise in 1910–1950, with 0.10% average annual growth rate. Between 1910 and 1950, a slight increase in the cropland area was experienced and the cropland area reached 6.71 million ha. (4) Gradual decrease in 1950–1999, with -0.26% average annual growth rate. After 1950, the cropland area declined and dropped to 5.90 million ha in 1999, constituting 6.71% of the land area.

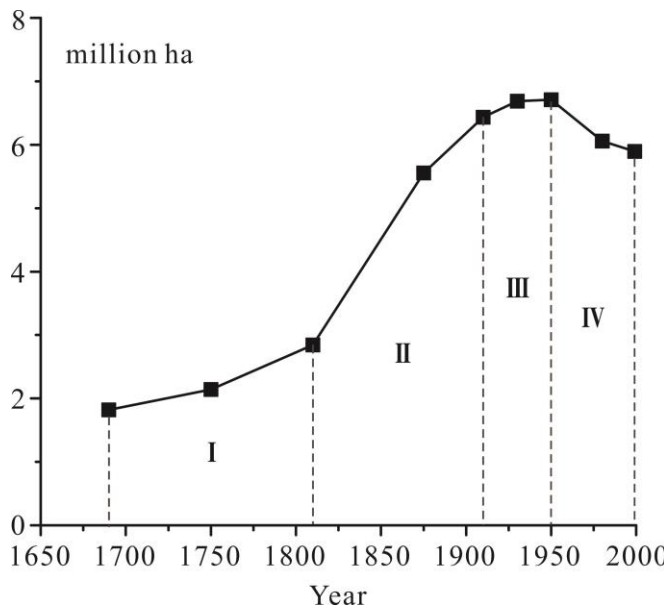

Figure 4 Changes in the total cropland area in Scandinavia during 1690–1999

## 4.2 Changes at the country level

For Sweden, the cropland area was just 0.68 million ha in 1690 and rose slowly to 1.21 million ha in 1810. After 1810, Sweden had the largest cropland area in Scandinavia, growing from 2.89 million ha in 1875 to around 3.60 million ha in 1910. The next 40 years witnessed a slight growth in cropland area, reaching a peak in 1950 of 3.66 million ha and representing 52.57% of the total cropland area in Sweden. Then, the cropland area declined to 2.75 million ha by 1999.

For the first three time points, Denmark's cropland area grew gradually and accounted for 58.53%, 59.44%, and 52.17% of all cropland area in 1690, 1750, and 1810, respectively. The cropland area continued to rise at an average annual growth rate of 0.36% until 1950 when it reached its highest area of 2.68 million ha. After that, the figure slowly dropped to 2.63 million ha in 1999.

The cropland area difference between Sweden and Denmark narrowed after 1980. Compared with Sweden and Denmark, Norway had the smallest cropland area, which comprised only 6% of Scandinavia's total cropland area, on average. After a slow increase from 1690 to 1810, there was noticeable growth between 1810 and 1910. During 1690–1910 the cropland area increased by 4.5 times. Despite some slight fluctuations, the cropland area rose from 0.35

million ha in 1910 to 0.65 million ha in 1999, with an average annual growth rate of 0.70% (Figure 5).

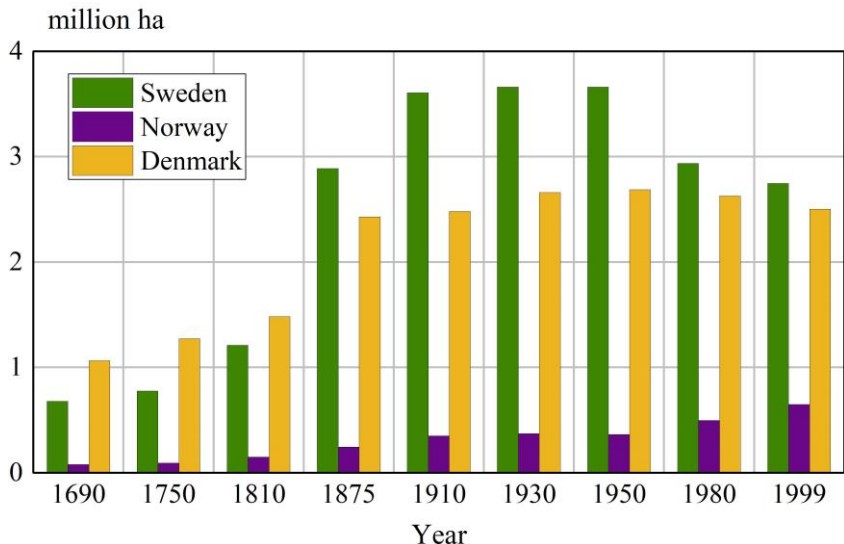

Figure 5 Changes in the total cropland area during 1690–1999 in Sweden, Norway and Denmark

## 4.3 Spatial cropland distribution patterns

The spatial patterns of cropland distribution from 1690 to 1999 in Scandinavia are shown in Figure 6. Denmark and southern Sweden already had extensive cropland cover in 1690, which reflected their long historical agricultural practices. Before 1810, cropland cover expanded in southern Scandinavia and remained stable in the north. Grid cells with more than 5% cropland accounted for 9.79% and 12.58% of the total grids cells in 1690 and 1810, respectively, while the number of grid cells with more than 60% cropland increased approximately four times. After 1810, cropland in northern Scandinavia experienced slight expansion, and the cropland fractions increased rapidly. In 1910, grid cells with more than 5% cropland represented a proportion of around 17.05%, while the percentage of grid cells with more than 60% cropland grew to 6%. During 1910–1950, cropland area changed gradually; cropland proportion changes in most grid cells were between -20% and 20%. After 1950, cropland areas began to decrease in most regions, especially in eastern Scandinavia. In western Scandinavia, the cropland area increased gradually and expanded to the north. Grid cells with more than 60% cropland accounted for 5.53% of the total grid cells and 28.43% of the grid cells that had cropland in 1999.

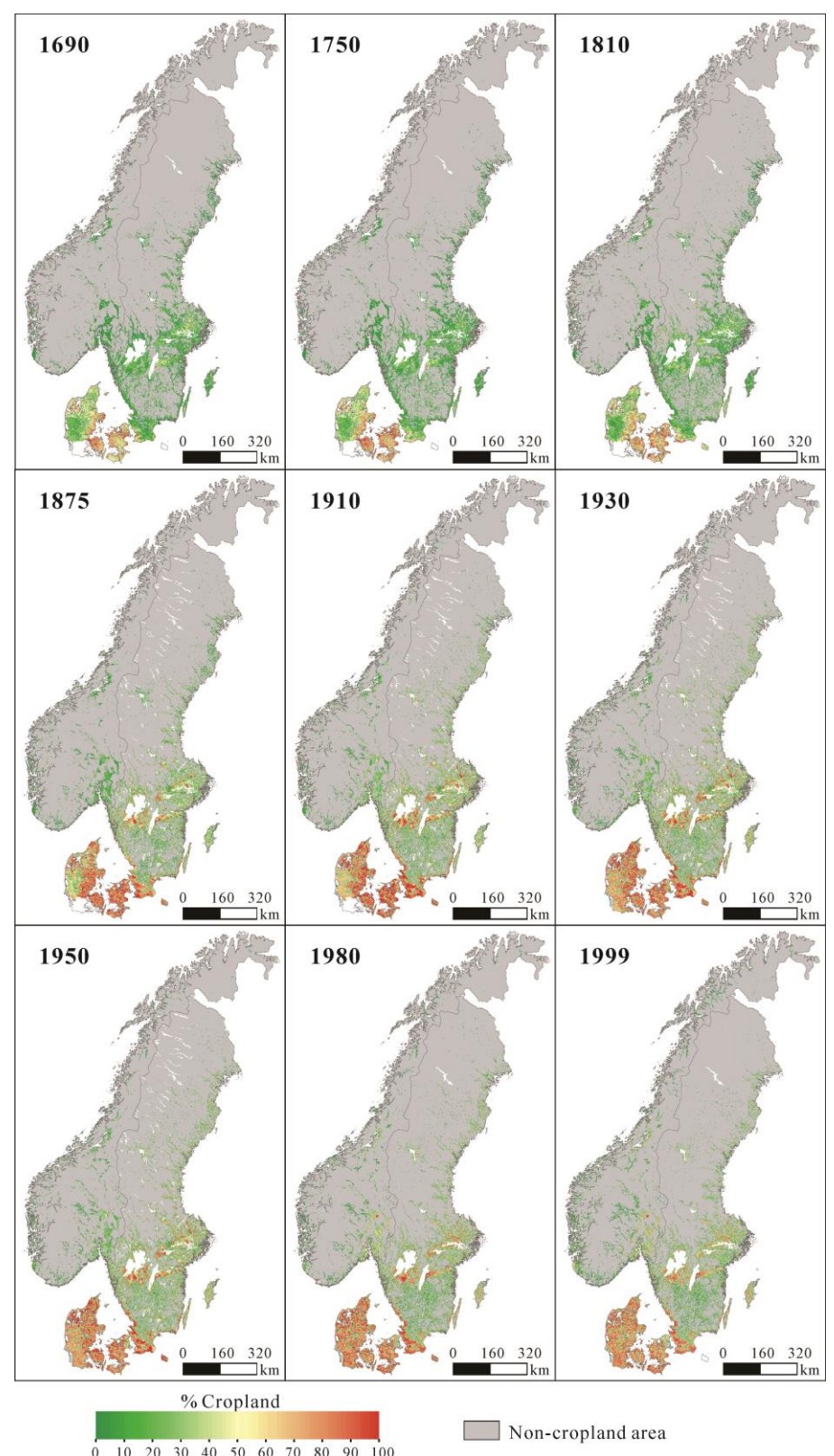

Figure 6 Spatial distribution of cropland area during 1690–1999 in Scandinavia

### 4.3.1 Sweden

In Sweden, about half of the area is covered by forest. Mountains, marshes, and lakes together cover approximately one third of the area. The cropland area has accounted for 1.50%–8.13% of Sweden's total land area over the past 300 years. In 1690, croplands were especially dense in southern Sweden, especially around the lakes of Vänern, Vättern, Mälaren and Hjälmaren as well as in Skåne, reflective of the long cultivation history. After that, the cropland spatial patterns became more intensive in southern Sweden and began to spread northward. Several grid cells with more than 80% cropland were observed and increased in 1910. During 1910–1950, spatial patterns of cropland distribution remained stable, except for slight increases in Västerbotten and Norrbotten counties in the north, and minor changes in Skåne (Malmöhus and Kristianstad counties) and Gotland. In the following years, the cropland area declined in most regions and the percentage of grid cells with more than 80% cropland dropped in 1980 and 1999. However, the coastal areas of Halland and Skåne still maintained high cropland fractions.

### 4.3.2 Norway

Mountains, forests, open heathlands, and grasslands dominate Norway's landscapes, and only about 3% of the land surface is suited for cultivation or arable farming (FAO, http://www.fao.org/family-farming/detail/en/c/358178/). In 1690, a small amount of cropland in Norway was distributed around the two agricultural centers, Olsofjorden and Trondheimsfjorden. From 1750 to 1810, the cropland spatial patterns around the two fjords expanded and cropland appeared in Nordland County in northern Norway. Then, the cropland fraction increased in the two agricultural centers during 1810–1910, and cropland began to appear in the northernmost county, Finnmark. Cropland in the two agricultural centers stabilized and even decreased but increased in other regions from 1910 to 1950. After 1950, cropland around Olsofjorden and Trondheimsfjorden started growing again. The dramatic growth occurred in the southwestern area of Rogaland. Cropland area in low-elevation coastal regions also increased.

### 4.3.3 Denmark

Denmark is among the most intensively cultivated countries in Europe. The long land cultivation history has included widespread cropland cover since 1690 in Denmark. Most high-fraction grid cells were distributed in eastern Denmark as soil conditions were more suitable for crop planting than in the western part of the country. The period between 1690 and 1810 experienced gradual growth, and the most rapid cropland increase occurred in eastern Denmark. During 1810 and 1910, a sharp increase in cropland area in both eastern and western Denmark was observed. Grid cells with more than 20% and 60% cropland accounted for 83.42% and 71.24% of the total grid cells in 1910. Since 1910, negative rates of change in cropland areas have dominated in northern and eastern Denmark, and positive rates in the western part, including Ribe, Ringkjøbing and Viborg counties. However, the changes in cropland areas of most grid cells were less than 20%. Although cropland areas in South Jutland, Ribe, Ringkjøbing, Viborg, and North Jutland counties increased moderately, those in other Danish counties declined from 1950 to 1999.

## 5 Discussion

### 5.1 Validation of the dataset developed in this study

To validate the newly created dataset, we used the satellite-based modern cropland cover data. Two sets of global land cover maps at 30-m resolution, from the Global Food Security Analysis-Support Data at 30 Meters (GFSAD30, https://croplands.org/app/map?lat=0&lng=0&zoom=2) Project and GlobeLand30 maps were found. However, GFSAD30 provides land cover map at 30-m resolution only for 2015 whereas GlobeLand30 has 30-m resolution maps in 2000, 2010, and 2020. Although there are differences among satellite-based maps of multiple time points in modern times, compared to the past 300 years, the period of 1985–2000 can be regarded as one time point. Thus, we choose the GlobeLand30 map for 2000 to validate our 1999 cropland dataset.

The images for land cover development classification and update of GlobeLand30 were mainly 30-m multispectral images, including TM5 ETM+ and OLI multispectral images from Landsat (USA) and HJ-1 (China Environment and Disaster Reduction Satellite). GlobeLand30 includes ten land-cover classes in total (Chen, et al., 2014); namely cultivated

land, forest, grassland, shrubland, wetland, water bodies, tundra, artificial surface, bare land, and perennial snow and ice. Cultivated land refers to the land used for cultivating crops. Paddy fields, irrigated upland, rainfed upland, vegetable land, cultivated pasture, greenhouse land, land mainly planted with crops and rarely with fruit trees or other trees, tea plantations, coffee plantations, and other economic croplands are included in this category (Chen, et al., 2014). However, cropland in our datasets only includes arable land (areas under temporary crops, temporary meadows and pastures, land temporarily fallow) and areas under permanent crops.

We compared the cropland area at the parish and county levels in 1999 in this study with cultivated land area from the 2000 GlobeLand30 map to validate our statistics. Then, we aggregated the 30-m resolution GlobeLand30 to 1 km and compared the result with our 1999 gridded cropland dataset to validate our allocation method. Comparing the cropland area at the parish and county levels in 1999 in this study with the cultivated land from the 2000 GlobeLand30 map shows that the cultivated land area from GlobeLand30 is 1.4 times that of our cropland area (Figure 7).

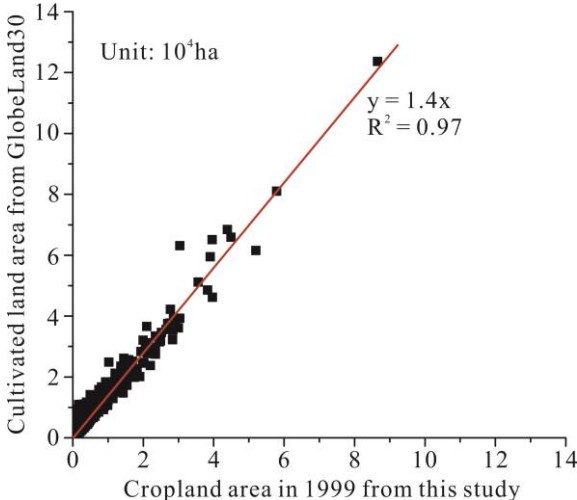

Figure 7 Comparison of the area at the parish and county levels between GlobeLand30 and this study

Comparison of the gridded cropland area at 1 km resolution between GlobeLand30 and this study shows grids with differences between -20% and 20% account for 64.81% of the total number of grids with the cropland area > 0 (Figure 8). Because cultivated pasture, greenhouse land, gardens, and so on are included in cultivated land from GlobeLand30, whereas only

arable land and permanent crops comprise cropland in this study, among all grids with cultivated land > 0, 79.44% of grids from GlobeLand30 have more cultivated land than this study.

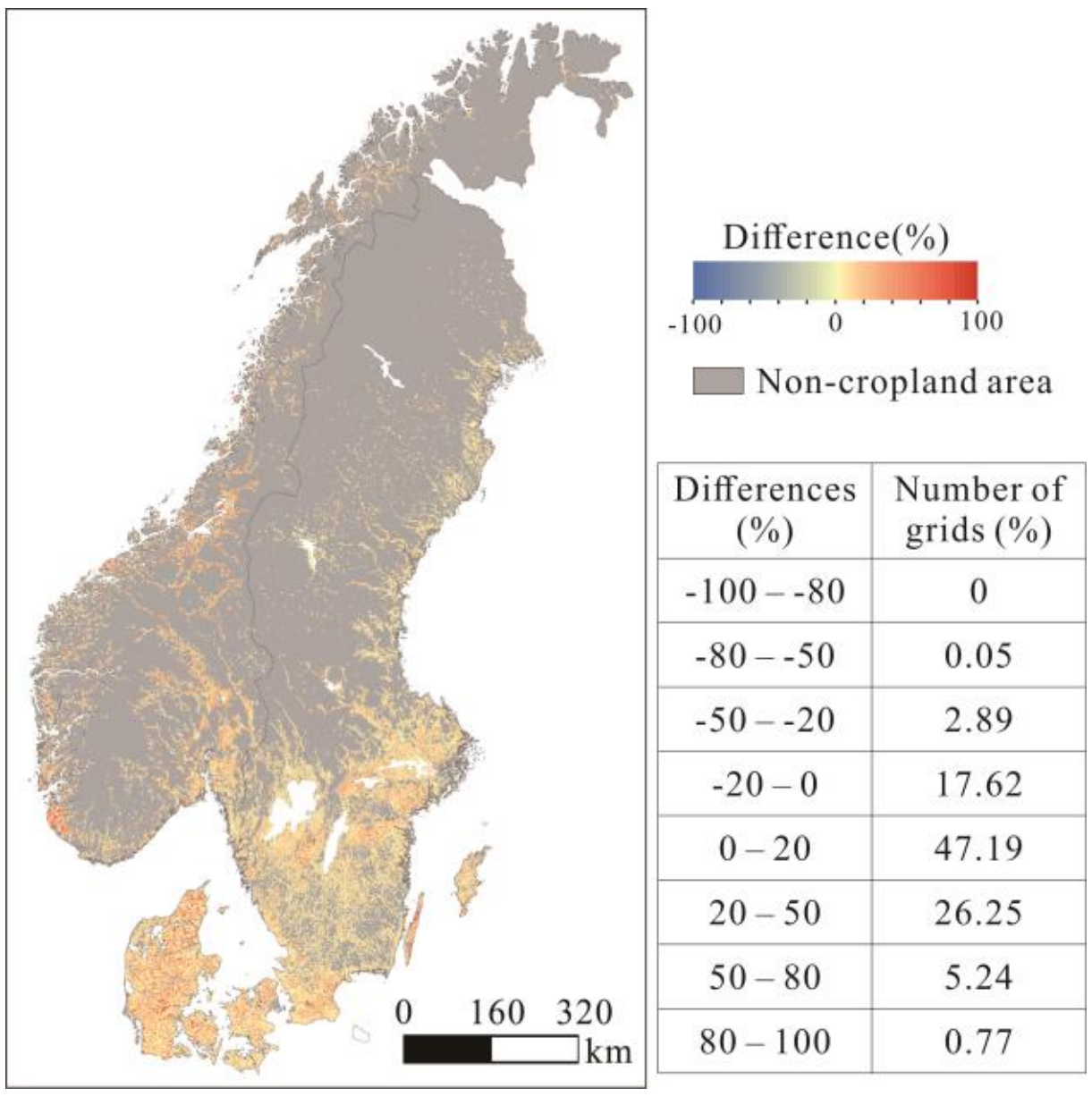

| Differences (%) | Number of grids (%) |
|---|---|
| -100 – -80 | 0 |
| -80 – -50 | 0.05 |
| -50 – -20 | 2.89 |
| -20 – 0 | 17.62 |
| 0 – 20 | 47.19 |
| 20 – 50 | 26.25 |
| 50 – 80 | 5.24 |
| 80 – 100 | 0.77 |

Figure 8 Comparison of the gridded cropland area between GlobeLand30 and this study

Because CLC2000 provides more agricultural (cultivated) land classes (Büttner, 2014), we used CLC2000 to further validate our cropland dataset. In Scandinavia, CLC2000 divides agricultural areas into four categories: “Arable land” (including one class, “Non-irrigated arable land”), “Permanent crops” (including one class, “Fruit trees and berry plantations”), “Heterogeneous agricultural areas” (including two classes, “Complex cultivation patterns” and “Land principally occupied by agriculture, with significant areas of natural vegetation”), and “Pastures”. Comparison of the gridded cultivated land area and agricultural areas at 1-km resolution between GlobeLand30 and CLC2000 shows the consistency of these two datasets. We compared the total area of “Arable land,” “Permanent crops,” and “Complex cultivation patterns” from CLC2000 with our cropland area for each parish and county (Figure 9). Figure 9 shows that the area from CLC2000 is 1.17 times that of our cropland area. Removing some marginal roads and natural lands included in “Arable land,” “Permanent crops,” and “Complex cultivation patterns” from CLC2000, our cropland areas at the parish and county levels are close to those from CLC2000.

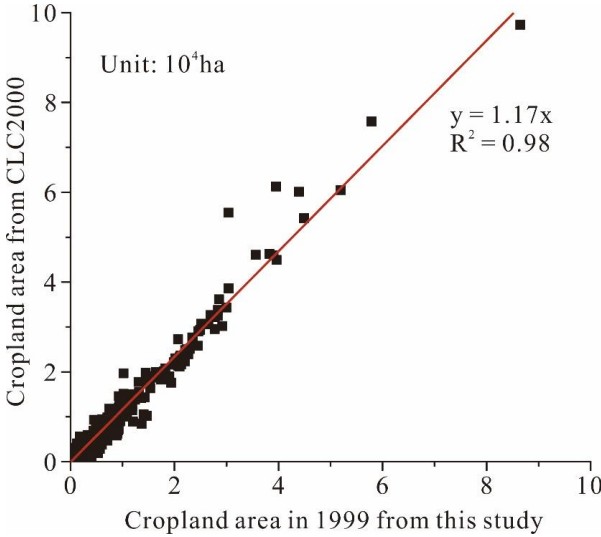

Figure 9 Comparison of the cropland area at the parish and county levels between CLC2000

and this study

Comparison of the gridded cropland area at 1-km resolution between CLC2000 and this study

shows grids with differences between -20% and 20% account for 90.54% of the total number

of grids with the cropland area > 0 (Figure 10). Our gridded cropland area is close to that in

CLC2000 overall, which indicates the reliability of our cropland dataset. However, around

Stockholm and Trondheim, in southeastern Norway and northern Denmark, respectively, our

cropland areas are slightly less than those from CLC2000.

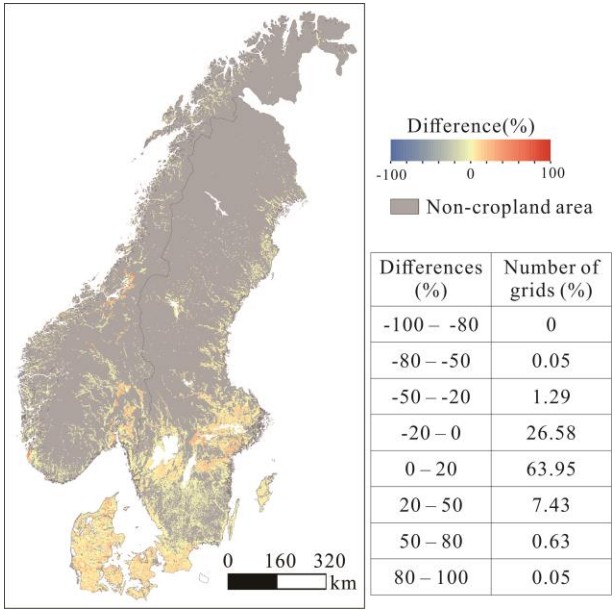

Figure 10 Comparison of the gridded cropland area between CLC2000 and this study

We also compared the total country level cropland area with that from previously published

studies. Because many studies of historical agriculture in Norway and Denmark were also

statistically based, cropland areas before 1980 were available only in Sweden. This study's cropland area is similar to the data from previous studies ($R^2$ = 0.967, slope = 1.05), which indicates the reliability of the results in this study (Figure 11). Because the studies of Anderberg (1991), Lindstad (2002) and Groth, et al. (1998) provided the cultivated land area but not the cropland area, their data show slightly larger areas than the data observed in this study.

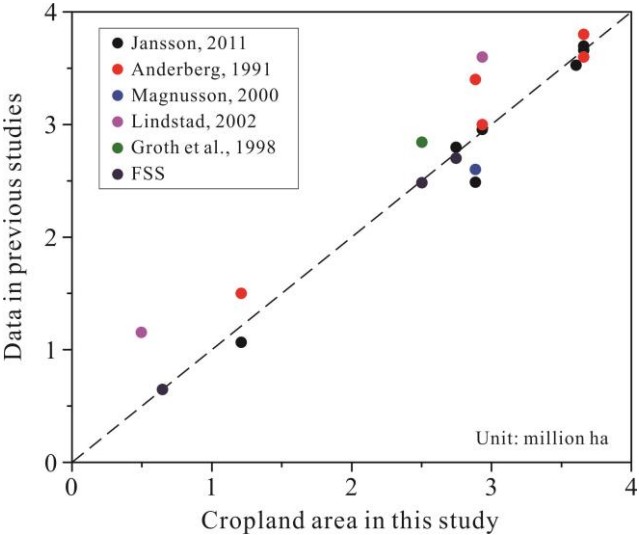

Figure 11 Comparison of the cropland data from previous studies and this study

Several missing data were interpolated in Denmark in 1688, 1750, 1800, and 1881 using the cropland fractions of their neighbors and the linear interpolation method. These interpolation methods were also used by Ramankutty and Foley (1999), Ye, et al. (2015), Wei, et al. (2016); He, et al. (2017), Li, et al. (2018), and Yu and Lu (2018); however, their interpolated data did not reduce the credibility of their datasets. As satellite-based data and survey data at the parish level were unavailable from 1688 to 1881, the interpolated data were impossible to validate using direct fitted observations. Based on the study of Fang et al. (2020), three methods could be used to assess the credibility of historical land cover datasets, including accuracy assessment (quantitative assessment based on quantitatively reconstructed regional land cover data), rationality assessment (qualitative assessment, including the regional historical facts-based rationality assessment and the expertise-based rationality assessment), and likelihood assessment (the credibility of the land cover data for given spatial or temporal units is inferred according to the degree of consistency in land cover data extracted from multiple

datasets). Because apart from the data sources we used, other quantitatively reconstructed regional land cover data in Denmark from 1688 to 1881 were unavailable, we employed a regional historical facts-based rationality assessment to analyze the reliability of our interpolated data in Denmark. The following Danish history suggests that linearly interpolated data are reasonable. The national tax system stipulated that each household in Denmark had 50 acres of cropland, no more and no less. The population of Denmark grew steadily from 1690 to 1881. During this period, wars did not cause sudden changes in cropland area in Denmark. The agricultural reform that began in 1789 changed the relationship between landlords and tenant farmers but did not cause a sudden change in cropland area (Jespersen, 2018).

## 5.2 Comparison with global datasets

To show the improvement made by our dataset, three widely used global datasets HYDE 3.2 (Klein Goldewijk et al., 2017), PJ (Pongratz et al., 2007), and KK10 (Kaplan et al., 2011) were selected to compare with our results. As KK10 only provided the sum of cropland and pasture area data, it was much larger than the total cropland area from other studies (Figure 12). For Scandinavia, the total cropland area from this study was between that from the HYDE 3.2 and PJ results. However, differences are apparent at the country level, especially in Norway and Denmark. For Sweden, though cropland areas were close to those from different datasets before 1810, the PJ results were far less than those from HYDE 3.2 and this study after 1875.

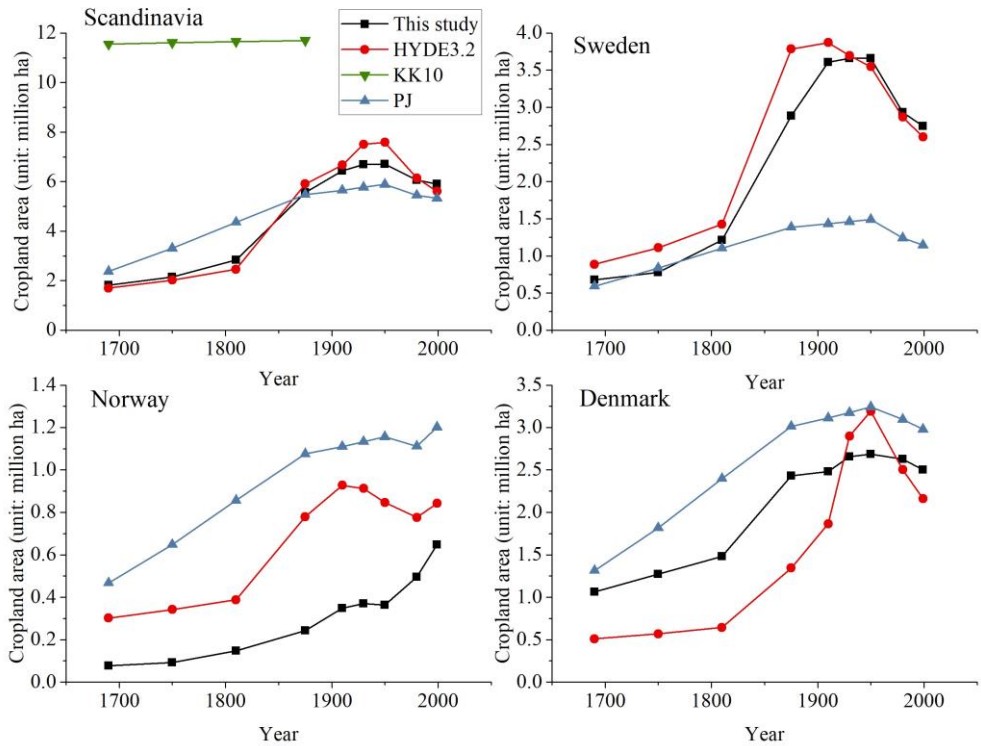

Figure 12 Comparison of the total cropland area among HYDE3.2, PJ, KK10, and this study
Compared with HYDE 3.2, large differences were found during 1690–1875 in Sweden and
Denmark, and at all time points in Norway. Although the RD between PJ and this study was
close before 1810 in Sweden and after 1875 in Denmark, it was even larger than that between
HYDE 3.2 and this study in Norway. PJ datasets overestimated the total cropland area in
Norway and Denmark, and after 1875 in Sweden. HYDE 3.2 also overestimated the total
cropland area in Norway but underestimated it before 1910 and after 1980 in Denmark,
compared with this study (Table 3). Different definitions of cropland in Norway caused
cropland area overestimation in Norway by HYDE 3.2 and PJ. Based on the FSS data, we
inferred that some permanent meadow area was included in the cropland land area by HYDE
3.2 and PJ.

Table 3 Relative difference ratio (RD, unit: %) between global datasets and this study

| Time points | Sweden | | Norway | | Denmark | |
|---|---|---|---|---|---|---|
| | RD(HYDE3.2) | RD(PJ) | RD(HYDE3.2) | RD(PJ) | RD(HYDE3.2) | RD(PJ) |
| 1690 | 30.81 | -12.51 | 290.74 | 504.51 | -52.04 | 23.59 |
| 1750 | 42.97 | 7.67 | 269.80 | 599.98 | -55.40 | 42.73 |
| 1810 | 17.84 | -8.89 | 162.50 | 479.01 | -56.62 | 61.85 |

| 1875 | 31.16 | -51.98 | 220.43 | 342.24 | -44.67 | 24.08 |
| 1910 | 7.40 | -60.32 | 166.00 | 218.16 | -24.79 | 25.42 |
| 1930 | 1.02 | -60.10 | 146.19 | 205.73 | 9.00 | 19.44 |
| 1950 | -3.12 | -59.30 | 132.50 | 217.46 | 18.83 | 20.67 |
| 1980 | -2.32 | -57.71 | 56.38 | 124.03 | -4.67 | 17.88 |
| 1999 | -5.25 | -58.33 | 30.04 | 85.42 | -13.59 | 19.06 |

We selected time points with RDs < 30% to compare spatial patterns of cropland distribution from PJ, HYDE 3.2, and this study. PJ had a spatial resolution of 0.5° × 0.5°, but HYDE 3.2's resolution was 5′ × 5′. To reduce the resampling error, cropland area data from this study were aggregated to 0.5° × 0.5° and 10 km × 10 km to compare with PJ and the resampled HYDE 3.2, respectively. For Sweden, although the differences in total cropland area during 1690–1810 from PJ and this study were small, the spatial patterns of cropland distribution varied substantially. Based on the PJ dataset, more cropland was shown in northern Sweden and Skåne, but the cropland area around lakes Vänern, Vättern, Mälaren, and Hjälmaren in southern Sweden was less than that in this study. For Denmark, more cropland area was allocated in western and southeastern Denmark by PJ, but cropland area in northern and southwestern was scarce (Figure 13), which is inconsistent with the facts. Because the land in eastern Denmark was more conducive to cultivation than that in the West, the eastern cropland area gradually decreased with urbanization development. Increased afforestation and the need for urban expansion steadily reduced the agricultural area in eastern Denmark (Pedersen and Møllenberg, 2017).

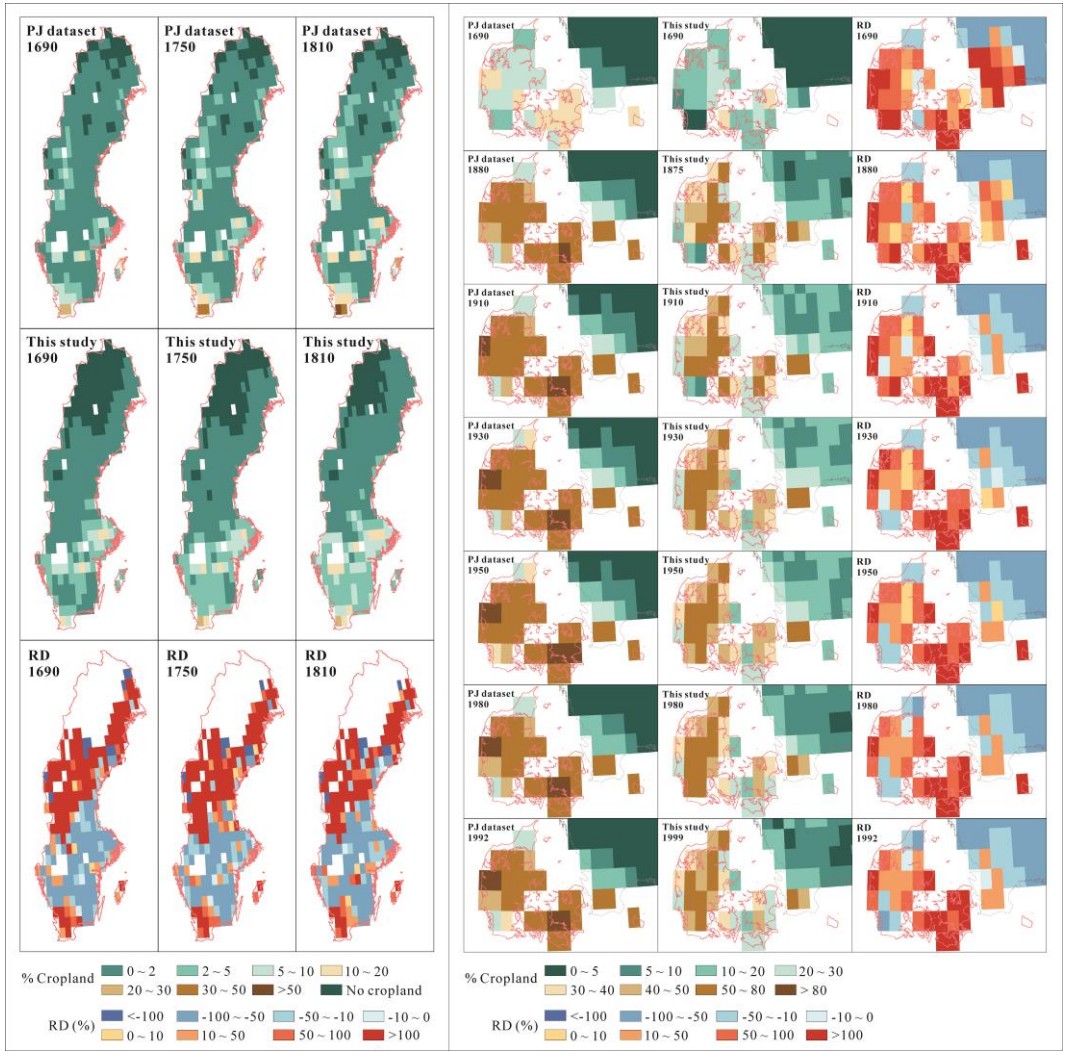

Figure 13 Comparison of the spatial distribution of cropland area between PJ and this study
(The left figure is the comparison in Sweden and the right one is the comparison in Denmark)

Although spatial cropland patterns were in agreement in Sweden between HYDE 3.2 and this

5    study, around the lakes Vänern, Vättern, Mälaren, Hjälmaren, and in the northeast coastal area,

there were significantly more cropland areas in HYDE 3.2 than in this study. However, in

Blekinge, Hallands, Jönköping, Kalmar, and Kronoberg counties in southern Sweden, there

were fewer cropland areas in HYDE 3.2 than in this study. Moreover, no cropland area was

allocated in northern Sweden from HYDE 3.2, but statistics show that this area has cropland.

10    The number of grid cells with RD more than 50% or less than -50% accounted for 53%–64%

of the total grid cells from 1810 to 1999 (Figure 14). There was little difference in the spatial

patterns of cropland from 1910 to 1999 for Denmark between HYDE 3.2 and this study

(Figure 15). RDs of most grid cells were between -50% and 50%. The number of grid cells

with RDs between -10%–10% accounted for approximately 40% of the total grid cells in 1980

and 1999 because Denmark has a small land area and most land has been cultivated intensely. Moreover, the topographical differences within Denmark are relatively small. Thus, although HYDE 3.2 used the national cropland area, the cropland area allocation errors were small in Denmark.

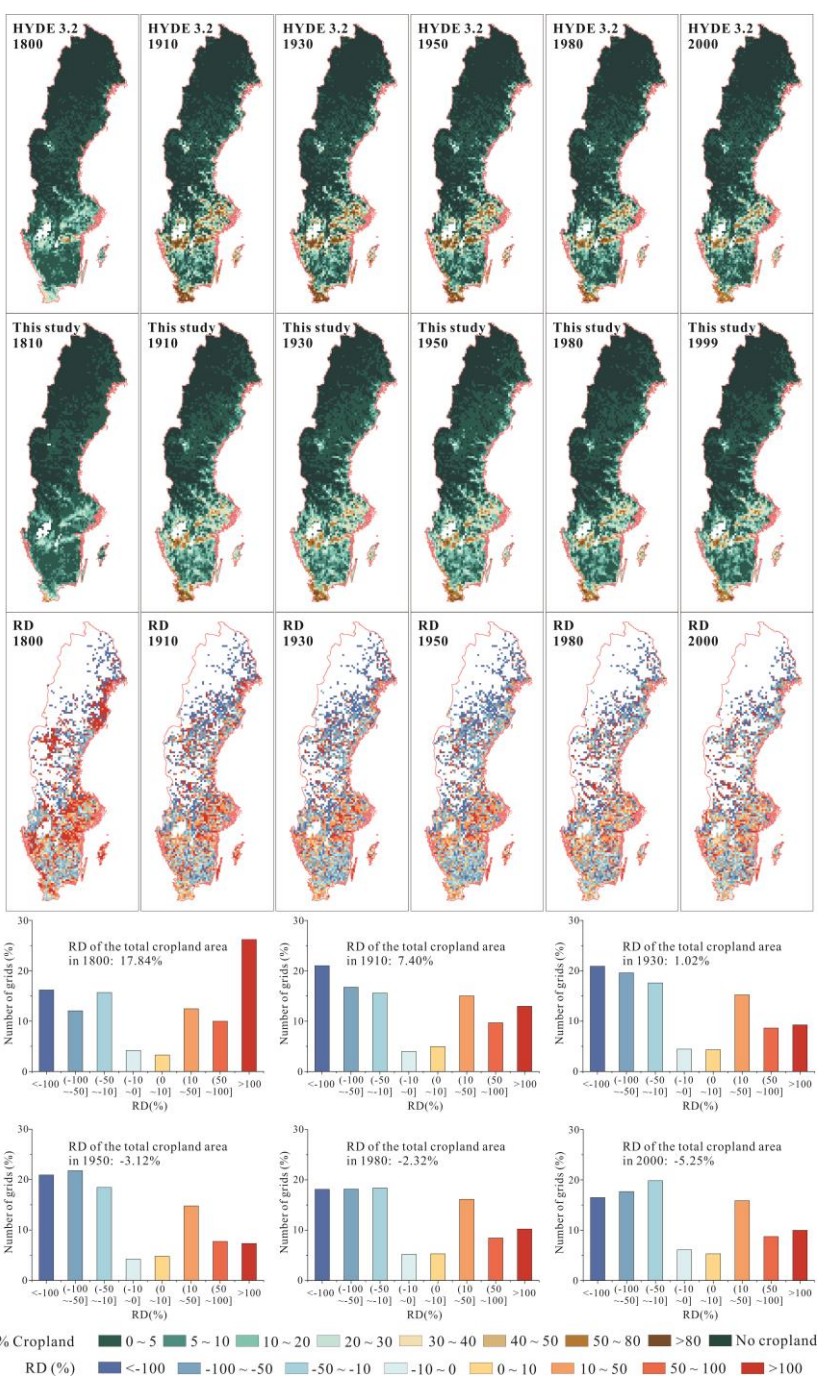

Figure 14 Comparison of the spatial distribution of cropland area between HYDE 3.2 and this study in Sweden

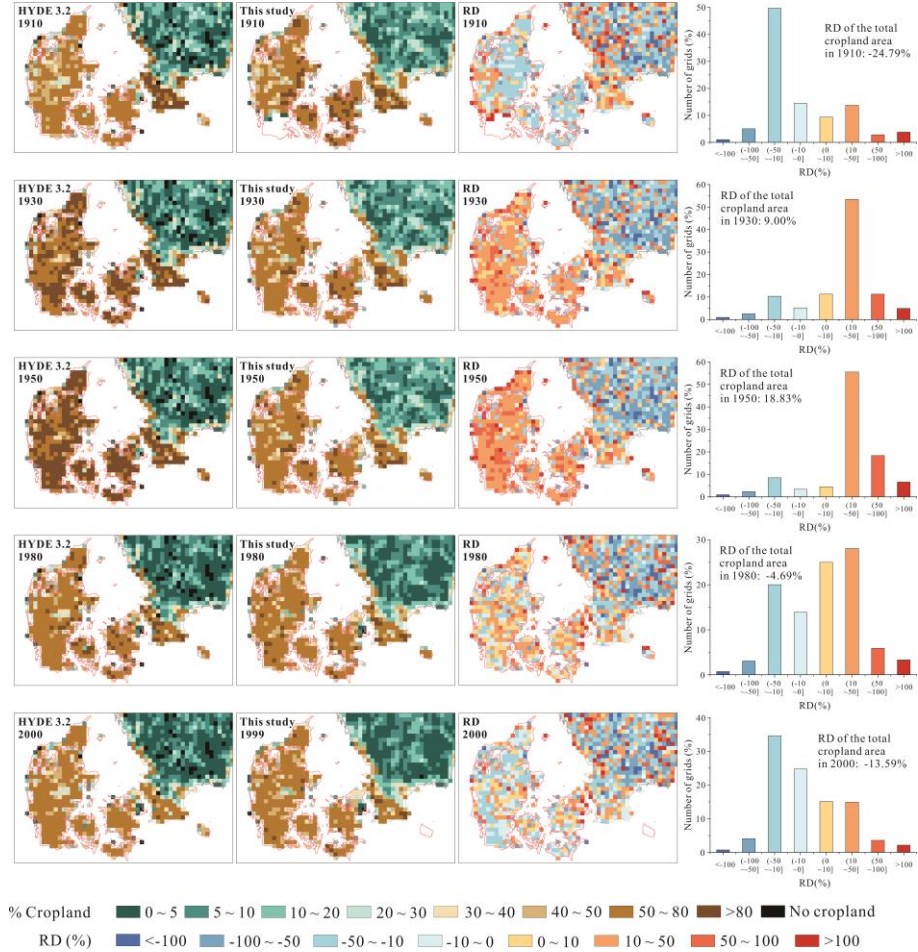

Figure 15 Comparison of the spatial distribution of cropland area between HYDE 3.2 and this study in Denmark

## 5.3 Uncertainties

5    To develop the cropland area dataset in Scandinavia from 1690 to 1999, we have used many methods to ensure data accuracy. However, there are still uncertainties in this work. Our cropland is defined as the sum of arable land and land under permanent crops. Arable land comprises areas under temporary crops, temporary meadows and pastures, and land temporarily fallow. Although the temporarily fallow and permanent crop areas account for a

10    small proportion of the total cropland area, the absence of records about the size of fallow land makes our cropland before 1875 in Sweden and Norway smaller than the real value. Moreover, we used elevation and slope as factors that affect the spatial distribution of cropland, which may allocate a smaller amount of cropland area to grids that should not have cropland historically and make the spatial patterns of cropland insufficiently concentrated.

Failure to use settlements in the cropland area allocation model also introduced errors in our gridded dataset.

## 6 Data availability

All cropland data cover for 1690, 1750, 1810, 1875, 1910, 1930, 1950, 1980, and 1999 at a spatial resolution of 1 km are available in https://doi.org/10.1594/PANGAEA.926591 (Wei et al., 2021).

## 7 Conclusions

Based on the collected cropland data of each administrative unit from statistics and previous studies and using a range of data processing and cropland area allocation methods, we developed the cropland area dataset at a spatial resolution of 1km in Scandinavia from 1690 to 1999. Our reconstruction indicated that the cropland area developed slowly before 1810, then increased rapidly until the beginning of the 20th century and remained stable for around 40 years before declining in 1999. At the country level, the cropland area change trends in Sweden and Denmark were almost identical to that in Scandinavia. The cropland areas of both Sweden and Denmark reached a peak in 1950. Norway had the least cropland area, which increased gradually from 1690 to 1999. The spatial patterns of cropland distribution showed that Denmark and southern Sweden already had extensive cropland cover in 1690. In the following 100 years, cropland expanded in southern Scandinavia and remained stable in the north. During 1910–1950 the cropland area changed slightly but began to decrease in southern and eastern Scandinavia after 1950.

The statistically based accuracy of our gridded cropland dataset has been validated by comparison with satellite-based data (GlobeLand30 and CLC2000) in 2000, although our cropland area is slightly smaller than that from satellite-based data. Comparing our dataset and global datasets shows that KK10 has much larger total cropland area than this study in Scandinavia. Although the total cropland area is in agreement in Scandinavia among HYDE 3.2, PJ and this study, more considerable differences are found in cropland areas at the country level. The differences in cropland spatial patterns are also non-negligible, even at time points where the total cropland area differences are small in Sweden and Denmark

between PJ and this study. HYDE 3.2 allocates more cropland area to highly cultivated regions in Sweden but has a lower cropland area between the highly cultivated regions in southern Sweden.

Although some cropland area allocation errors introduced uncertainties with our reconstruction, this study improved descriptions of historical cropland change in Scandinavia. Our cropland dataset is an essential reference for a better understanding of the complex climate system.

## Acknowledgments

We thank Peder Dam from Odense City Museums and Pia Frederiksen from Aarhus University for providing us cropland data of Denmark in 1688, 1800, 1881 and 1998, and Ulf Jansson for providing an administrative map of Sweden in 1750. This research is supported by National Key R&D Program of China (No. 2017YFA0603304), National Science Foundation of China (No. 41807433), Basic Research Program (Natural Science Foundation) of Jiangsu Province (No. K20180804), and the Fund of China Scholarship Council.

## Author Contribution

WX, WM and LB designed the work. WX wrote the manuscript. FX, YY and CT provided suggestions on structure and methods. ZC helped with downloading the data for cropland allocation. All the authors contributed to the review of the manuscript.

## Competing interests

The authors declare that they have no conflict of interest.

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
