# Peer review of "Dataset of 1-km cropland cover from 1690 to 1999 in Scandinavia"

_Earth System Science Data, 2020_

## Referee Comment (RC1) · Anonymous Referee #1 · 28 Aug 2020

General comments. The authors estimated cropland area of Scandinavia for 1690-2015 using historical documents and allocated them into grids with a resolution of 30 seconds for simulation of climatic effects of land cover change. They also compared their results with prior study: HYDE3.2. Overall, this work is complete and the produced dataset is accessible via the given identifier. However, some issues should be addressed further. The area estimation of historical cropland is more important than spatial allocation. Generally, there are many problems in historical records, which cannot be directly used without evaluation and correction. But in this manuscript, the reliability evaluation of the data used to estimate cropland area was missing. The methods of historical cropland estimation based on proxy data were not introduced in detail, and the estimation and allocation results were not validated or calibrated (comparison

with HYDE3.2 is not calibration). These problems will undermine the reliability of the dataset. Additionally, the results of this manuscript and HYDE3.2 are close (Figure 5a), so what's the advantages of this dataset? Although some differences exist in spatial pattern (Figure 6), the conclusion "the results of this study are better that HYDE3.2, and are more close to real land use history" cannot be reached by readers. Finally, the study area is small, and the proportion of cultivated land is also very small (The max value is about 7% in 1950). However, the dominated land cover type, forest, was excluded in this study, which greatly undermined the significance of this dataset.

Specific comments. Introduction. Why is it necessary to reconstruct the area and distribution of historical cropland in this area? What is the importance or necessity of reconstruction in this area? Area estimation. The reliability evaluation of the data used to estimate cropland area was missing. Spatial allocation. The allocation results were not validated or calibrated. Discussion. The results of this manuscript and HYDE3.2 are close (Figure 5a), so what's the advantages of this dataset? "Underestimation" and "overestimation" are inappropriate. In addition, it is suggested that the author should supplement the use of this dataset as widely as possible, particularly compared with previous datasets, including HYDE, so as to let readers understand the specific value of this dataset.

Technical corrections. Page 3, line 9, forestland is unavailable in SAGE dataset. Page 5, line 6, repeated "Li et al." Page 11, line 18-20, reference(s) are needed. Figure 4 is not indispensable. Page 21, line 15 "underestimated", and line 21 "overestimation" are inappropriate.
* * *

---

## Referee Comment (RC2) · Anonymous Referee #2 · 9 Sep 2020

This study has allocated cropland area in Scandinavia from 1690 to 2015 into 30-arc second grids using the available statistical dataset at administrative level. Later, this allocated grid data is compared with HYDE3.2 dataset in the same area. Dataset is downloadable and usable format. Development of high resolution and historical precise cropland dataset is necessary for several environmental and policy related studies. This study has provided some baseline with the available data collection from 1690 for the study area but there are several limitations observed in this study as authors move from county level dataset to grids based dataset. These limitation as briefly given as follow: 1. Major limitation of this study is the data uncertainty and gaps in the methodology. For data uncertainty – it can be observed that, this study did not perform any validation of the dataset and entire dataset is clearly based on only statistical datasets

in the region – therefore, use of satellite based dataset as explained in the introduction is irrelevant to the study and therefore its allocation to the girds is without base. 2. CORINE dataset used in the background of grids need further explanation as the allocation in the 1700 cannot be similar as allocation in 2010. 3. Another limitation is 'cropland definition' – meaning of croplands is not clearly given in the dataset as data is the mixture of grasslands, fallow lands and sometime cropland area is converted using the volume of seeds to area. 4. Result and discussion part mainly explain the changes in the croplands in the allocated croplands in the study area but author should provide more detailed results on the allocation on croplands itself, for example, how much is the error percent in the grids in each year or how the allocation is showing the granularity as compared to country level polygons, detailed statistical analysis on the allocation for uncertainty and area values. 5. Lastly, the paper need serious grammatical English correction and some restructuring for example, methodology can be well explained with flow charts and results may have first section to explain the plain allocated maps itself rather than the change in croplands. There are several places where writing can be improved. Below are few specific comments: Specific comments: • Line 10 in section 1 –"The decrease of natural vegetation is accompanied by an increase in cropland area." need to support with references. Decrease of natural vegetation may have other drivers including increasing agricultural activities. • This study introduction should be focused on agriculture in Scandinavia and should provide more background on it in the introduction rather than detailed explanation about global croplands and its changes over time. Authors may provide more literature review on croplands in Scandinavia and avoid exaggerated details about global croplands and its changes. • In methods, there are several gap in the information and analysis. For example, missing data of several counties for many time-stamps is calculated using interpolation but there is no any validation performed to support the output. Also, the allocation of croplands from county-level historical dataset to grids is not clearly explained and uncertainty in the conversion process remains firm. • Methods did not explain the accuracy and validation of the resulting cropland dataset and therefore,

the reliability and usage of this dataset is limited. • High resolution dataset term is leading in the entire paper – 30 arc second or 0.5 degree datasets may not be considered high resolution. Author may need to rethink on the use of the term high resolution. • Lastly, it is very important to know why Scandinavia region for this study and then why Agricultural lands for historical land use study if there can be another significant land covers which may affect. The context of the paper need to be updated with better explanation and focus on the main goal of the paper. • Also, the goal of the study is misleading as it is different in the introduction line 26 in Section 1 vs in methodology in line 4 in section 3. • Overall the work is significant with further additions and analysis along with English corrections.
* * *

---

## Author Comment (AC1) · 31 Jan 2021

**Response to referee comments**

**Anonymous Referee #1**

1, Generally, there are many problems in historical records, which cannot be directly used without evaluation and correction. But in this manuscript, the reliability evaluation of the data used to estimate cropland area was missing.

→We used two types of data sources in this study, statistics and previous studies. From the beginning of the 19th century, statistics were allowed to cover more and more social areas. Over time, the data became increasingly reliable, and the process was completed in the early 20th century (Linde and Palm, 2014). Thus, we need to validate the statistics before the 20th century. Cropland data in 1665, 1723, and 1809 for Norway, and cropland data in 1907, 1936, 1950, and 1980 for Denmark were from statistics. The rest of the cropland area data came from previous studies. All cropland data from previous studies we used was also based on statistics. The cropland area data before the 20th century from earlier studies was calibrated using regional historical maps (Linde and Palm, 2014; Dam and Jakobsen, 2008; Odgaard and Rømer, 2009). For cropland area data during 1875-1999 in Sweden and Norway, Li et al. (2013) used statistics directly, and the cropland data were smaller than the real values before the 20th century. Due to the historical census's unavailability, we compared the cropland area data in 1999 from Li et al. (2013) with that in 2000 from Eurostat agricultural census. Eurostat collects information on the structural characteristics of the agricultural holdings, including land use every 10 years as an agricultural census, with two or three additional, intermediate sample surveys carried out in-between in Norway. The difference was found between statistics and census in Norway, and we calibrated the cropland area data in Norway from 1690 to 1999 (See **Page 10, lines 2-19; Page 11, lines 1-9 in revised manuscript**).

2, The methods of historical cropland estimation based on proxy data were not introduced in detail, and the estimation and allocation results were not validated or calibrated (comparison with HYDE3.2 is not calibration). These problems will undermine the reliability of the dataset.

→We have added more information on the assessment and calibration of the data sources. Allocation method validation was also added. Because there was no real value of cropland area in each 1km × 1km grid from 1690 to 1999 in Scandinavia, the allocation results calibration was impossible to achieve. Therefore, we chose Non-irrigated arable land in 2000 from CORINE Land Cover to validate our allocation method (See **3 Methods**).

3, Additionally, the results of this manuscript and HYDE3.2 are close (Figure 5a), so what's the advantages of this dataset? Although some differences exist in spatial pattern (Figure 6), the conclusion "the results of this study are better that HYDE3.2, and are more close to real land use history" cannot be reached

by readers.

→ HYDE 3.2 works with a global model with assumptions that are supposed to be valid worldwide. However, historical realities differ from these assumptions. Examples are shown in China (Fang et al., 2020), the U.S. (Yu and Lu, 2018), and the European part of Tsarist Russia (Zhao et al., 2020). The differences between HYDE 3.2 and regional results are also different. Because the total cropland area in Scandinavia is much smaller than China, the U.S., and Tsarist Russia, the absolute difference between HYDE 3.2 and this study is also small. But the relative differences were between -13% and 13%. The bigger differences are shown on the national scale. The relative differences between HYDE 3.2 and this study are -5% ~ 43%, 30% ~ 291% and -57% ~ 19% in Sweden, Norway and Denmark, respectively. Norway's cropland area accounts for only 10% of the total cropland area in Scandinavia. Although Norway has the largest relative difference, it has little effect on relative differences in Scandinavia's total cropland area. Besides, HYDE 3.2 is larger than this study in Sweden but smaller in Denmark before 1930, making a small difference between HYDE 3.2 and this study in Scandinavia (See **5.1 Comparison with global datasets**).

Spatial differences between HYDE 3.2 and this study are critical to the accuracy of the cropland area dataset. In HYDE 3.2, the cropland area at the country level was allocated to grids based on population density, soil suitability, rivers, slope, and temperature. These factors determined whether and how much cropland area was allocated in each grid. This study allocated the cropland area of each Parish/Municipality/County to grids. For grids in administrative units with small cropland areas, it will not be allocated too much even if they have high weight for cropland allocation. In Scandinavia, spatial differences between HYDE 3.2 and this study showed that HYDE 3.2 model allocated cropland area to some grids in administrative units without cropland area based on this study. Or no cropland area was allocated by HYDE 3.2 model in all grids in administrative units with cropland area (See **Figure 11**). These spatial differences also reflect the necessity for spatial explicit historical cropland reconstructions in Scandinavia, although the total cropland area is tiny.

4, Finally, the study area is small, and the proportion of cultivated land is also very small (The max value is about 7% in 1950). However, the dominated land cover type, forest, was excluded in this study, which greatly undermined the significance of this dataset.

→For spatial explicit land cover reconstructions, cropland area data is often the basis for reconstructing grassland and forest land. Especially for the spatial pattern change of forest land, knowing the spatial extent of cropland can determine where the original forest land has been reduced. Except for natural factors, cropland area changes are mainly driven by human factors. Human activities have great uncertainty, and it isn't easy to simulate with models at present. Scandinavia has more detailed historical cropland area statistics than forest records. At some time points, even cropland areas at the parish level are available, which is necessary for improving the accuracy of

cropland area allocation. Thus, the forest was excluded in this study.

Although the study area is small and the total cropland area is also very small in Scandinavia, the amount of data in this study is large since we collected very detailed cropland data (Please see Table 1 below). After cropland area allocation, there are about 0.8 million 1km × 1km grids with cropland area data for each time point. The total number of grids is more than 7 million.

Table 1 The amount of cropland area data (before allocation) in this study

| Time points | The number of administrative units with cropland area data |
|:---:|:---:|
| 1690 | 3010 |
| 1750 | 3000 |
| 1810 | 4575 |
| 1875 | 2566 |
| 1910 | 982 |
| 1930 | 1100 |
| 1950 | 1104 |
| 1980 | 1013 |
| 1999 | 2570 |

5, Why is it necessary to reconstruct the area and distribution of historical cropland in this area? What is the importance or necessity of reconstruction in this area?

→There are several reasons that we choose Scandinavia as our study area. Although Scandinavia's total cropland area is less than the area of forest land, Scandinavia has a long history of land cultivation. Farmers were drivers of economic and social change for a long history. Over the past few centuries, agriculture in Scandinavia has undergone significant changes. Studies on agricultural history in Scandinavia mainly concentrated on agricultural policy, agricultural economy, settlement and population, and landscape history. But the spatial explicit historical land cover dataset of Scandinavia was scarce. The reconstruction of historical farmland depends on the richness and quality of data sources. Scandinavia has good historical cropland area data at the Parish/Municipality/County level. Time points with cropland area records are close in Sweden, Norway and Denmark, which helps us reconstruct the cropland area change from 1690 to 1999. We have explained the necessity to reconstruct the historical cropland area change in Scandinavia in the Introduction (See **Page 3, lines 26-29, Page 4, and Page 5, lines 1-6**).

6, The reliability evaluation of the data used to estimate cropland area was missing.

→We have added the assessment and calibration of our used cropland area data (See **Page 10, lines 2-19, Page 3, lines 1-9**). The newly developed dataset is also validated (See **5.1 Validation of the dataset developed in this study**).

7, The allocation results were not validated or calibrated.

→We have validated the allocation method (See **3.4 Allocation method validation**).

8, The results of this manuscript and HYDE3.2 are close (Figure 5a), so what's the advantages of this dataset? "Underestimation" and "overestimation" are inappropriate.

→ This question is the same as question 3. We have answered it.

9, In addition, it is suggested that the author should supplement the use of this dataset as widely as possible, particularly compared with previous datasets, including HYDE, so as to let readers understand the specific value of this dataset.

→ We have added comparisons with PJ and KK10 datasets. The differences were also analyzed (See **5.1 Comparison with global datasets**).

10, Technical corrections. Page 3, line 9, forestland is unavailable in SAGE dataset. Page 5, line 6, repeated "Li et al." Page 11, line 18-20, reference(s) are needed. Figure 4 is not indispensable. Page 21, line 15 "underestimated", and line 21 "overestimation" are inappropriate.

→We have corrected your mentioned errors. We used "underestimated" and "overestimation" to show the difference between HYDE 3.2 and this study. We have modified the sentences more clearly.

**Anonymous Referee #2**

1, Major limitation of this study is the data uncertainty and gaps in the methodology. For data uncertainty – it can be observed that, this study did not perform any validation of the dataset and entire dataset is clearly based on only statistical datasets in the region – therefore, use of satellite based dataset as explained in the introduction is irrelevant to the study and therefore its allocation to the girds is without base.

→Besides the statistics, data sources in this study include the results from previous studies. We have added the explanation of our used datasets. These data are also validated and calibrated (See **Page 10, lines 2-19, Page 3, lines 1-9**). We used the satellite-based dataset in cropland area allocation, and the maximum cropland extents for allocation is based on the satellite-based dataset. We also gave more information about the use of satellite-based dataset (See **Page 14, lines 2-25, Page 15, lines 1-8**).

2, CORINE dataset used in the background of grids need further explanation as the allocation in the 1700 cannot be similar as allocation in 2010.

→Our first version of the manuscript used The 300m CCI-LC maps developed by European Space Agency (ESA), but not CORINE dataset. Thanks for your advice and we found land cover classes from CORINE dataset are more than those from ESA,

which are more suitable for cropland area allocation in Scandinavia. We allocated the cropland area to grid cells considering different time points and administrative units. The maximum cropland extent maps used in different time points were different in our revised manuscript. Based on the total cropland areas in different administrative units, we allocated the cropland area to the extents of "Arable land, Permanent crops, Discontinuous urban fabric", "Complex cultivation patterns", "Land principally occupied by agriculture, with significant areas of natural vegetation", "Pastures, Artificial, non-agricultural vegetated areas, Industrial, commercial and transport units" and "Forests within 1km of arable land and pastures" from CORINE datasets in turn, until all the cropland was allocated (See **Page 14, lines 2-25, Page 15, lines 1-8**).

3, Another limitation is 'cropland definition' – meaning of croplands is not clearly given in the dataset as data is the mixture of grasslands, fallow lands and sometime cropland area is converted using the volume of seeds to area.
→ The category "Cropland" defined by FAO (http://www.fao.org/) was used for the cropland in this study. The "Cropland" includes areas under temporary crops, temporary meadows and pastures, land with temporary fallow, and permanent crops. The collect cropland area data was calibrated to meet the definition of "Cropland" used in this study. However, Sweden and Norway before 1875 had only cropland data recorded as the volume of seed, so we converted the volume of seed records to cropland area, which belongs to "Cropland" can fill the gap in cropland area before 1875.

4, Result and discussion part mainly explain the changes in the croplands in the allocated croplands in the study area but author should provide more detailed results on the allocation on croplands itself, for example, how much is the error percent in the grids in each year or how the allocation is showing the granularity as compared to country level polygons, detailed statistical analysis on the allocation for uncertainty and area values.
→Due to the lack of "true value" of the spatial explicit cropland area in history, it is almost impossible to analyze the uncertainty of our cropland area allocation results. However, we selected arable land area in 2000 as an example to analyze the uncertainty (See **3.4 Allocation method validation**).

5, Lastly, the paper need serious grammatical English correction and some restructuring for example, methodology can be well explained with flow charts and results may have first section to explain the plain allocated maps itself rather than the change in croplands.
→We have added flow charts for explaining our methods. The first section was also added to explain the allocated maps. A professional person has polished the manuscript and serious grammatical English has been corrected.

Specific comments:
6, Line 10 in section 1 –"The decrease of natural vegetation is accompanied by

an increase in cropland area." need to support with references. Decrease of natural vegetation may have other drivers including increasing agricultural activities.

→ "The decrease of natural vegetation is accompanied by an increase in cropland area." was concluded based on the result of Pongratz et al. (2008) (See **Page 2, lines 8-10**). The expression was maybe misleading. We have deleted this sentence.

7, This study introduction should be focused on agriculture in Scandinavia and should provide more background on it in the introduction rather than detailed explanation about global croplands and its changes over time. Authors may provide more literature review on croplands in Scandinavia and avoid exaggerated details about global croplands and its changes.

→ One of this study's primary purposes is to verify and improve the global land use datasets using regional historical materials with higher resolution. Thus, we must introduce the advantage and the shortage of global datasets. We have revised the "Introduction", and the redundant parts were removed. According to your advice, we introduced the agricultural and land use history in Scandinavia in the "Introduction", please see **Page 3, lines 26-29, Page 4, lines 1-19**.

8, In methods, there are several gap in the information and analysis. For example, missing data of several counties for many time-stamps is calculated using interpolation but there is no any validation performed to support the output. Also, the allocation of croplands from county-level historical dataset to grids is not clearly explained and uncertainty in the conversion process remains firm.

→ Interpolation of missing cropland data based on cropland in neighboring counties or cropland change trends at adjacent time points is the two most commonly used data interpolation methods for historical cropland reconstruction. Since only Denmark had 3% missing data, interpolation results using different methods have little effect on Denmark's total cropland area. We have given more detailed explanations about cropland area allocation (See **3.3 Cropland area allocation**). The cropland allocation method was also validated (See **3.4 Allocation method validation**).

9, Methods did not explain the accuracy and validation of the resulting cropland dataset and therefore, the reliability and usage of this dataset is limited.

→ We have added the validation of our dataset (See **5.1 Validation of the dataset developed in this study**).

10, High resolution dataset term is leading in the entire paper – 30 arc second or 0.5 degree datasets may not be considered high resolution. Author may need to rethink on the use of the term high resolution.

→ Compared with the satellite-based data in modern times, 30 arc seconds (we have changed the resolution to 1km in the revised manuscript) is not high resolution.

However, the current global cropland area from HYDE 3.2 has the highest resolution of 5 minutes. At present, the highest resolution of the existing historical cropland cover dataset for Scandinavia is 5′×5′, also from HYDE 3.2. So for the historical cropland area dataset in Scandinavia, 1km is a high resolution. Moreover, since the 1km×1km resolution cropland dataset is obtained by allocating the cropland area of each administrative unit, further increasing the spatial resolution will lead to a decrease in the accuracy of the dataset.

11, Lastly, it is very important to know why Scandinavia region for this study and then why Agricultural lands for historical land use study if there can be another significant land covers which may affect. The context of the paper need to be updated with better explanation and focus on the main goal of the paper.

→For spatial explicit land cover reconstructions, cropland area data is often the basis for reconstructing grassland and forest land. Especially for the spatial pattern change of forest land, knowing the spatial extent of cropland can determine where the original forest land has been reduced. Except for natural factors, cropland area changes are mainly driven by human factors. Human activities have great uncertainty, and it isn't easy to simulate with models at present. Scandinavia has more detailed historical cropland area statistics than forest records. At some time points, even cropland areas at the parish level are available, which is necessary for improving the accuracy of cropland area allocation. Thus, the forest was excluded in this study. We have added the explanations about why we chose Scandinavia and "Cropland". Please see the **Introduction.**

12, Also, the goal of the study is misleading as it is different in the introduction line 26 in Section 1 vs in methodology in line 4 in section 3.
→ We have re-write the **Introduction** and **Methods**.

13, Overall the work is significant with further additions and analysis along with English corrections.
→ We have added more explanations and polished our manuscript.

**References:**

Linde, M., Palm, and L. A.: Sverige 1810: Befolkning, jordbruk, skog, jordägande, Rapport för Vetenskapsrådets project Databasen Sverige 1570-1805: befolkning, jordbruk, jordägande, Institutionen för historiska studier, Göteborgs universitet, 2014.

Dam, P., and Jakobsen, J. G. G.: Atlas over Danmark: Historisk-Geografisk Atlas, København: Det Kongelige Danske Geografiske Selskab, 2008.

Odgaard, B., and Rømer, J. R.: Danske Landbrugs-landskaber gennem 2000 år, Gylling: Narayana Press, 2009.

Fang, X., Zhao, W., Zhang, C., Zhang, D., Wei, X., Qiu, W., and Ye, Y.: Methodology for credibility

assessment of historical global LUCC datasets, Science China Earth Sciences, https://doi.org/10.1007/s11430-019-9555-3, 2020.

Yu, Z., and Lu, C.: Historical cropland expansion and abandonment in the continental U.S. during 1850 to 2016, Global Ecology and Biogeography, 27, 322-333, https://doi.org/10.1111/geb.12697, 2018.

Zhao, Z., Fang, X., Ye, Y., Zhang, C., and Zhang, D.: Reconstruction of cropland area in the European part of Tsarist Russia from 1696 to 1914 based on historical documents, Journal of Geographical Sciences, 30(8), 1307-1324, https://doi.org/10.1007/s11442-020-1783-y, 2020.

---

## Referee Report (RR1)

My brief comment on each given comment is as follows:

1. My previous comment of major limitation about data validation and calibration is not explained satisfactory. In the revision, author had compared past statistical or other study dataset to compare their datasets at the grid level but that comparison do not add any value because the dataset developed by author has used baseline of the statistical data to allocate each value into grids. Since the baseline data used for mapping has itself used for validation (in this case areal comparisons at administrative level), which cannot be necessarily qualify validation. Thus, major limitation of this study still stands.

2. This study has mentioned that they have used CCI-LC maps of 2000, but revised paper has maps of 1690 to 1999, thus this study does not use any satellite data and has reorganized and spatially allocated the historical dataset where no validation is available.
To provide suggestion on validation, the developed dataset may not needed to validate for all the past years but if author can validate dataset using the satellite data maps from satellite such as Landsat where map can be developed at 30m resolution from 1980 to 1999 or at least from 1985 with some limitations of data and compare these maps with the allocated maps developed by this study: those results may be some kind of comparison and provide the base to validate the results of this study for some years at least.

3. As observed in this study, cropland data is collected from several studies and different governments : thus do not hold a single cropland definition and need further explanation. This study did not provide clear explanation on it. How did author combine all this datasets when cropland definitions of different dataset were different. What was the basis, how did it affect the fusion?

4. As I provided above solution about using those satellite dataset available in historical years like Landsat is available from 1985 , which can be used to check spatially accuracy and allocation for precision of maps. I suggest authors to implement this method rather than not validating the results and providing blind spatial allocations with no base.

5. This work still needed high level of English correction and organization of writing. For example, in revised version, discussion section has lot of methodological details and results and very less discussion. This paper has lot of scope to work on organizing the sections and restructuring the paper while providing English corrections.

6. This data is not high resolution maps: Author may call it spatial maps as previous dataset just have county level details but using high-resolution is not suitable. In remote sensing terms less than 10m pixel can be considered as high resolution according to definitions provided by several international research organizations such as USDA, UN,FAO.

7. Data reliability is still questionable as the validation and gap filling is not explained or analyzed properly. Although interpolation is the only way to gap fill data but the interpolated data need to be validate for further use.

---

## Referee Report (RR2)

I have carefully reviewed the major revision and have enlisted below replies to each comment.

1.  As per my suggestion, author has removed 'high resolution' term from the title: I am aware that Author has developed dataset which might be high resolution for the given timeline but by the remote sensing definition, it is still not high resolution and term proves to be unsuitable. Thus, I appreciate author decision to skip the term high resolution.

2.  Author has successfully implemented my comment on validating data using currently available dataset using CLC dataset. To move forward, I would really like to see the comparison with SAGE cropland dataset as well as it was approximately developed using similar methods and similar timeline. I understand author has performed comparisons using HYDE dataset but SAGE dataset comparison would be more suitable.

3.  Overall, author has sincerely worked on my comments and implemented the suggestions satisfactorily and I think this paper has improved significantly after it.

I have minor suggestions in the structure of paper as below:

1.  In introduction, I would suggest to reduce introduction on global cropland mapping importance and focus on why it is important to provide cropland maps in your study area. There is background on agriculture in the study area but it is not clearly suggesting why grid based cropland mapping is necessary. I would suggest author to add a papragraph on it and reduce the other part focusing on global cropland mapping little bit.
    Second, I am not getting the story in continuous manned while reading the introduction. My suggestion would be rearrange some information to get the flow of context.

2.   You can remove the sub-divisions in data sources section. It just creating confusing and misreading the reader about usage of satellite data and other datasets. You have used some reference datasets and some statistical dataset, which can come under one hood of " available cropland data from the study area".

3.  In methods, I would suggest just to have one flowchart covering all the steps rather than having many flowcharts in each section. In that way, it will be easier for reader to understand entire methods in one go. I am happy to review the new one flowchart if required.

4.  Validation should go under results section. Also, time-series changes should be under one section and not two subsections if possible.

5. Discussion should discuss the limitations of the dataset clearly.

Overall, I am very satisfied with the author with all the changes. The above minor suggestions would hopefully help the readability of the manuscript.

---

## Editor Decision (ED1)

Abstract okay except various acronyms not defined.

Page 2 iine 9: IPCC uses the familiar acronym LULCC. You should adopt the same or specify how and why yours differs from the expected community term. Check all uses of LUCC vs LULCC.

Page 2 line 11: Usually, instead of citing the entire WG I report of AR-5, authors cite a specific chapter or even a page number?

Page 2 line 21: List of acronyms (e.g. SAGE, HYDE, PJ and KK10) need definition. Given the range of acronyms used in this manuscript, from old Swedish sources to modern satellite products, authors should consider a table of acronyms with definitions as an Appendix?

Page 2 line 29: If Le Quéré et al., 2018 is supposed to represent most recent global carbon budget, more recent versions exist (e.g. Friedlingstein et al. 2020) exist.

Page 3 lines 1through 5 "There uncertainties were unneglectable in regional applications" ?? Following sentence adds to confusion rather than clarifying. This means that uncertainties acceptable in global context become too large in regional products? "There" or 'their'? Confusing. I think you mean that assumptions made in global products become unacceptably large at regional contexts? But, for IPCC at least, most LULCC and AFOLU estimates come from most-recent national reports of varying quality and reporting date? If you want to declare a need to validate more carefully on regional scales for historical cropland changes, you have not made the point clearly.

Page 3 line 7: ALCC - what's this? Not defined. Same as AFOLU in IPCC terms? Or do you mean 'anthropogenic land-cover change' ala PAGES. If different, how and why justified?

Page 3 line 8: PAGELandCover6k mostly focuses on paleoclimate indicators (e.g. pollen) and not exclusively on regional patterns. Here you focus on small region (Scandinavia) with unusually-good historical records? How does this work fit with PAGES paleoclimate projects?

Page 3 line 10: "Errors" in regional reconstructions or in global products. Need clarity here.

Page 3 line 16: farmers are were, please make careful and consistent use of past tense.

Page 3 line 30 to page 4 line 1: "importance …. could fail to be determined precisely" What? Confusing!

Page 4 lines 15 to 19 - finally, a clear statement of intent. This text could replace much of what precedes it. Dataset will provide? Better: dataset provides!

Overall, good helpful description but methods, data and results sections need careful scrutiny and occasional re-write!

Page 36, around line 30: Reference list not in alphabetical order. Please check entire reference list for similar errors.

Typesetters and proofreaders from Copernicus will apply very careful very good language services for this manuscript but they will have many questions! Two changes suggested here: careful reading and re-writing by a native English speaker and careful definition of all acronyms (consider a list of acronyms as suggested) will make their job easier and your product better!

---

## Author Response (AR2)

**Response to referee comments**

General reply. Thank you for your insight comments which have improved our work greatly. The object of this study is to develop a longer (compared to the work of Li et al. (2013)) historical gridded cropland dataset over the period of 1690–1999 at the resolution of 1 km. Although in the ear of remote sensing, land cover data could reach a resolution of 30 m (such as GlobeLand30), our dataset has the highest resolution over the historical period of 1690–1970 at least. We have deleted "high-resolution" in this revised version in accordance with your advice. Besides traditional allocation methods, remote sensing based CORINE Land Cover (CLC, https://land.copernicus.eu/pan-european/corine-land-cover) data is used as a reference in the cropland area allocation. We used CLC instead of CCI-LC map because CLC has agricultural land cover in eleven classes whereas CCI-LC map has only five classes. The reasonableness of our dataset is further explained using multiple independent methods. In this revision, modifications were carefully made.

My brief comment on each given comment is as follows:
1. My previous comment of major limitation about data validation and calibration is not explained satisfactory. In the revision, author had compared past statistical or other study dataset to compare their datasets at the grid level but that comparison do not add any value because the dataset developed by author has used baseline of the statistical data to allocate each value into grids. Since the baseline data used for mapping has itself used for validation (in this case areal comparisons at administrative level), which cannot be necessarily qualify validation. Thus, major limitation of this study still stands.

Reply: Thank you for your comments. We agree that the validation in this study is insufficient. However, validation is always the most difficult part because the reconstruction of historical datasets as they have no direct fitted observations. The actual past land cover data (referred to as the "true value") that serves as the credibility assessment baseline is not directly accessible and needs to be reconstructed in most cases. However, historical and natural records available for land cover reconstruction are very limited, and a widely accepted method for such an assessment remains to be developed (Fang et al., 2020). Therefore, to demonstrate the data production reasonableness, cross-comparison with other independent datasets is the most common approach (Yu et al., 2018; Zhao et al., 2020). The reasonableness of our cropland data before 1980 was investigated using other regional historical works (Anderberg, 1991; Groth et al., 1998; Magnusson, 2000; Lindstad, 2002; Jansson, 2011; FSS, https://ec.europa.eu/) in the previous revision.

In this revision, the reasonableness of our 1999 cropland data was validated using satellite-based land cover datasets (CLC and GlobeLand30 map, http://www.globallandcover.com/ ) from 2000. More explanations about the validation of our 1999 cropland dataset are listed in the responses to question 2.

We must explain that the comparison with global datasets is not intended to validate our datasets. One purpose of this study is to produce a historical cropland dataset based on cropland area at the **parish/municipality/county** levels, and to use our dataset to assess

and improve the global cropland dataset, which is based on cropland area allocation at the **national** level. Reconstructed regional land cover data derived from historical records are regarded as the baseline in most existing studies of credibility assessments of historical global land cover data (Fang et al., 2020). In the first version of our manuscript, we only compared our dataset with the most widely used global dataset HYDE 3.2. Subsequently, following referee #1's advice, we compared our dataset with additional global datasets in the previous revision.

2. This study has mentioned that they have used CCI-LC maps of 2000, but revised paper has maps of 1690 to 1999, thus this study does not use any satellite data and has reorganized and spatially allocated the historical dataset where no validation is available. To provide suggestion on validation, the developed dataset may not needed to validate for all the past years but if author can validate dataset using the satellite data maps from satellite such as **Landsat** where map can be developed at **30m resolution** from 1980 to 1999 or at least from 1985 with some limitations of data and compare these maps with the allocated maps developed by this study: those results may be some kind of comparison and provide the base to validate the results of this study for some years at least.

Reply: Thank you for your comments. As you suggested, we have tried our best to collect all the public available datasets we can access.

In the first version of our manuscript, we used CCI-LC map in 2000 as reference for cropland area allocation. According to your comments, we found that CORINE Land Cover (CLC) had more detailed information about the cropland area in Scandinavia. Thus, CLC data were applied to our cropland area allocation model, because CLC had agricultural land cover in **eleven** classes whereas CCI-LC map had only **five** classes (Table 1).

Table 1 Classes of agricultural areas from CLC-LC maps and CLC maps

| CCI-LC | Cropland, rainfed (Herbaceous cover) | Cropland, rainfed (Tree or shrub cover) | Cropland, irrigated or post-flooding | Mosaic cropland (>50%) / natural vegetation (tree, shrub, herbaceous cover) (<50%) | Mosaic natural vegetation (tree, shrub, herbaceous cover) (>50%) / cropland (<50%) | |
|---|---|---|---|---|---|---|
| CLC | Non-irrigated arable land | Permanently irrigated land | Rice fields | Vineyards | Fruit trees and berry plantations | Olive groves |
| | Pastures | Annual crops associated with permanent crops | Complex cultivation patterns | Land principally occupied by agriculture, with significant areas of natural vegetation | Agro-forestry areas | |

The CLC inventory was initiated in 1985 and updates have been produced in 2000, 2006, 2012, and 2018. However, CLC maps are only available after 2000 for Scandinavia. Thus, we used 2000 CLC map in this study. CLC2000 is produced by many countries in Europe by visual interpretation of high-resolution satellite imagery from **Landsat-7**

**ETM** (https://land.copernicus.eu/pan-european/corine-land-cover).

The CLC map of 2000 plays an essential role in our allocation methods; however, this was not explained clearly in the previous version. In this revision, modifications were made in the data and methods sections. **Please check the noted Sections (2.3 Satellite-based data, 3.2 Cropland area allocation into grid cells).**

According to your advice, to validate our cropland dataset from 1980 to 1999, we find that there are two global land cover maps at 30m resolution, Global Food Security Analysis-Support Data at 30 Meters (GFSAD30, https://croplands.org/app/map?lat=0&lng=0&zoom=2) Project and GlobeLand30 maps. However, GFSAD30 provides a land cover map at 30-m resolution only in 2015. GlobeLand30 has maps at 30-m resolution in 2000, 2010, and 2020. **Although there are differences between satellite-based maps of multiple time points in modern times, compared to the history over the past 300 years, 1985–2000 can be regarded as one time point. Thus, we chose the GlobeLand30 map from 2000 to validate our 1999 cropland dataset.**

[revised manuscript text omitted]

Figure 5 Comparison of the gridded cropland area between CLC2000 and this study

**Please check the Section 5.1 Validation of the dataset developed in this study.**

3. As observed in this study, cropland data is collected from several studies and different governments : thus **do not hold a single cropland definition** and need further explanation. This study did not provide clear explanation on it. How did author combine all this datasets when cropland definitions of different dataset were different. What was the basis, how did it affect the fusion?

Reply: Thank you for your comments. We regret not providing a clear explanation of our cropland definition. Table 2 is used here to give a clear explanation of the cropland definitions for all datasets. The category "Cropland" defined by the FAO (http://www.fao.org/) was used in this study. Thus, cropland in this study includes areas under temporary crops (A), temporary meadows and pastures (B), land temporarily fallow (C), and areas under permanent crops (D).

Statistics of all countries in Scandinavia recorded land use areas of all crops, temporary meadows and pastures and fallow land. Therefore, we selected the land areas classified as cropland defined by the FAO and calculated their total area. In Norway, statistics only provide the area of A. Based on the census from the Farm Structure Survey (FSS, https://ec.europa.eu/) in 2000 and 2010, the total size of fallow land and land under permanent crops accounted for approximately 0.9% of all cropland area. Thus, we used the total areas under temporary crops (A) as cropland area before 1810 in Norway.

For datasets from previous studies, authors gave clear cropland definitions. **In Sweden**, "*åker*" in studies from SND (https://snd.gu.se/en/catalogue/study/SND0910) and Li et al. (2013) included land under temporary crops (A), land under temporary meadows and pastures (B), and temporarily fallow land (C). Because the census from FSS shows D accounts for only 0.1% of the total cropland area, we used the total area of A, B, and C as the cropland area. **In Norway**, Li et al. (2013) used the total area of A, B, C, D, and E (Permanent grassland and meadow) as the cropland area. We identified their data sources (NSD kommunedatabase, https://kdb.nsd.no/kdbbin/kdb_start.exe) and re-collected A, B, C, and D as cropland. **In Denmark**, both "ager" in the dataset of Dam and Jakobsen (2008) and "agerjord" in the dataset of Odgaard and Rømer (2009) indicate the total of A, B, and C. Because the census from FSS shows land area under permanent crops (D) accounted for approximately 0.4%–1% in Denmark, we used the total of A, B, and C as cropland.

We have explained the definitions of cropland from different sources more clearly. Please check the indicated sections (**2.1 Cropland data and 3.1.1 Cropland data collection and calibration**).

Tabel 2 Cropland definitions of the data sources

| Data sources | Spatial coverage | Years | Reference | Cropland definitions (Categories included in recorded cropland) | Combination |
|---|---|---|---|---|---|
| | | | | | |

| | | | | | |
|---|---|---|---|---|---|
| *Sockenvis jordbruksstatistik* | Sweden | 1690, 1750, 1810 | SND | A, B, C (Åker) | Census from FSS shows the area of D accounted for about 0.1% of the total cropland area in Sweden. We use the total of A, B and C as cropland. |
| *Statistiske studier over folkemængde og jordbrug i Norges* | Norway | 1665, 1723 | Aschehoug, 1890 | A | Census from FSS shows the total size of B, C, D accounted for about 0.9% of the total cropland area in Norway. We use A as cropland. |
| *Historisk Tidsskrift* | Norway | 1809 | Hovland, 1978 | A | |
| *Atlas over Denmark: Historisk-Geografisk Atlas* | Denmark | 1688 | Dam and Jakobsen, 2008 | A, B, C (Ager) | Census from FSS shows the area of D accounted for about 0.4%~1% of the total cropland area in Denmark. We use the total of A, B and C as cropland. |
| *Danske landbrugs-landskaber gennem 2000 år* | Denmark | 1800, 1881, 1998 | Odgaard and Rømer, 2009 | A, B, C (Agerjord) | |
| *Statistisk Aarbog 1912* | Denmark | 1907 | Danmarks Statistik, 1912 | A, B, C, D | We use the total of A, B, C and D as cropland |
| *Statistiske Meddelelser 1936, 1950 and 1980* | Denmark | 1936, 1950, 1980 | Danmarks Statistik, 1936, 1950 and 1980 | A, B, C, D | |
| *Cropland in Scandinavian Peninsula* | Sweden, Norway | 1875, 1910, 1930, 1950, 1980, 1999 | Li et al., 2013 | Norway: A, B, C, D, E | |
| | | | | Sweden: A, B, C (Åker) | Census from FSS shows the area of D accounted for about 0.1% of the total cropland area in Sweden. We use the total of A, B and C as cropland. |

Notes: A—Areas under temporary crops; B—Areas under temporary meadows and pastures; C—Land with temporary fallow; D—Areas under permanent crops; E—Permanent grassland and meadow

4. As I provided above solution about using those satellite dataset available in historical

years like Landsat is available from 1985 , which can be used to check spatially accuracy and allocation for precision of maps. I suggest authors to implement this method rather than not validating the results and providing blind spatial allocations with no base.

Reply: This comment is explained in the responses to question 2.

5. This work still needed high level of English correction and organization of writing. For example, in revised version, discussion section has lot of methodological details and results and very less discussion. This paper has lot of scope to work on organizing the sections and restructuring the paper while providing English corrections.

Reply: Thank you for your comments. We have moved the methodological details to **Method** Section and added datasets validation details in **Discussion** Section. We also restructured the paper. Our revised manuscript has been edited by Elsevier Language Editing Services.

6. This data is not high resolution maps: Author may call it spatial maps as previous dataset just have county level details but using high-resolution is not suitable. In remote sensing terms less than 10m pixel can be considered as high resolution according to definitions provided by several international research organizations such as USDA, UN,FAO.

Reply: Thank you for your comments. We have deleted "high-resolution" in the latest revised version in accordance with your comments.

7. Data reliability is still questionable as the validation and gap filling is not explained or analyzed properly. Although interpolation is the only way to gap fill data but the interpolated data need to be validate for further use.

Reply: Thank you for your comments. We have validated our cropland dataset using GlobeLand30 and CLC2000, please check the responses to question 2.

[revised manuscript text omitted]

---

## Author Response (AR3)

**Response to referee comments**

General reply. Thank you for your insight comments which have improved our work greatly. To show the improvement made by our regional dataset, three widely used global datasets HYDE 3.2, PJ, and KK10 were selected to compare with our results. SAGE cropland dataset was not selected because PJ dataset was developed based on SAGE. The cropland data source in PJ dataset for Scandinavia was from SAGE dataset over the period of 1700–1992. In this revision, we added comparison of the total cropland area between SAGE and this study. Because the cropland gridding method of PJ is improved compared to SAGE, we compared the spatial distribution of cropland area with PJ dataset but not SAGE dataset. We understand our timeline is similar with SAGE dataset, but the methods of SAGE are completely different from this study.

1. In introduction, I would suggest to reduce introduction on global cropland mapping importance and focus on why it is important to provide cropland maps in your study area. There is background on agriculture in the study area but it is not clearly suggesting why grid based cropland mapping is necessary. I would suggest author to add a papragraph on it and reduce the other part focusing on global cropland mapping little bit.
Second, I am not getting the story in continuous manned while reading the introduction. My suggestion would be rearrange some information to get the flow of context.
Reply: Thank you for your comments. We have rearranged the **Introduction** and made it more kind for readers. We also explained more about the necessities of historical grid-based cropland data in Scandinavia. Please check the pages 2–4.

2. You can remove the sub-divisions in data sources section. It just creating confusing and misreading the reader about usage of satellite data and other datasets. You have used some reference datasets and some statistical dataset, which can come under one hood of " available cropland data from the study area".
Reply: Thank you for your comments. We have removed the sub-divisions in data sources section.

3. In methods, I would suggest just to have one flowchart covering all the steps rather than having many flowcharts in each section. In that way, it will be easier for reader to understand entire methods in one go. I am happy to review the new one flowchart if required.
Reply: Thank you for your comments. In the last version, because Figure 3 includes a flow chart and a table, the combination of Figure 1–3 will increase the complexity of the methodology flowchart. Thus, we combined Figure 1 and Figure 2, but kept the flowchart of cropland area allocation model (Figure 3 in the last version, but Figure 2 in this version).

4. Validation should go under results section. Also, time-series changes should be under one section and not two subsections if possible.

Reply: Thank you for your comments. We have moved "Validation of the dataset developed in this study" to **Results** section. Time-series changes are under one section in this revision.

5. Discussion should discuss the limitations of the dataset clearly.

Reply: Thank you for your comments. We have discussed more about the uncertainties of our dataset. Please check "**5.2 Uncertainties**".

---

## Author Response (AR4)

Response to editor's comments

Abstract okay except various acronyms not defined.

Page 2 iine 9: IPCC uses the familiar acronym LULCC. You should adopt the same or specify how and why yours differs from the expected community term. Check all uses of LUCC vs LULCC.
Reply: Thank you for your comments. We have changed LUCC to LULCC.

Page 2 line 11: Usually, instead of citing the entire WG I report of AR-5, authors cite a specific chapter or even a page number?
Reply: Thank you for your comments. In the last version of our manuscript, we have cited the name of chapter. "**Anthropogenic and Natural Radiative Forcing**" is the chapter's name. In this version, we also cite the page number. Please see **page 38 lines 23-29**.
Reference: Myhre, G., Shindell, D., Bréon, F.-M., Collins, W., Fuglestvedt, J., Huang, J., Koch, D., Lamarque, J.-F., Lee, D., Mendoza, B., Nakajima, T., Robock, A., Stephens, G., Takemura, T., and Zhang, H.: **Anthropogenic and Natural Radiative Forcing**. In: Climate Change 2013: The Physical Science Basis. Contribution of Working Group I to the Fifth Assessment Report of the Intergovernmental Panel on Climate Change, [Stocker, T.F., Qin, D., Plattner, G.-K., Tignor, M., Allen, S.K., Boschung, J., Nauels, A., Xia, Y., Bex, V. and Midgley, P.M. (eds.)]. Cambridge University Press, Cambridge, United Kingdom and New York, NY, USA, **688pp.**, 2013.

Page 2 line 21: List of acronyms (e.g. SAGE, HYDE, PJ and KK10) need definition. Given the range of acronyms used in this manuscript, from old Swedish sources to modern satellite products, authors should consider a table of acronyms with definitions as an Appendix?
Reply: Thank you for your comments. We have added the definitions of SAGE, HYDE, PJ and KK10. We also add a table of acronyms with definitions as an Appendix.

Page 2 line 29: If Le Quéré et al., 2018 is supposed to represent most recent global carbon budget, more recent versions exist (e.g. Friedlingstein et al. 2020) exist.
Reply: Thank you for your comments. We have changed this reference to the most recent version of global carbon budget. Please see **page 2 line 29**.

Page 3 lines 1through 5 "There uncertainties were unneglectable in regional applications" ?? Following sentence adds to confusion rather than clarifying. This means that uncertainties acceptable in global context become too large in regional products? "There" or 'their'? Confusing. I think you mean that assumptions made in global products become unacceptably large at regional contexts? But, for IPCC at least, most LULCC and AFOLU estimates come from most-recent national reports of varying quality and reporting date? If you want to declare a need to validate more carefully on regional scales for historical cropland changes, you have not made the point clearly.
Reply: Thank you for your comments. We are sorry there is a mistake in the sentence "There uncertainties were unneglectable in regional applications". We have deleted this sentence and given more detail explanation of the uncertainties of global datasets. Please see **page 3 lines 1-16**.

Page 3 line 7: ALCC - what's this? Not defined. Same as AFOLU in IPCC terms? Or do you mean

'anthropogenic land-cover change' ala PAGES. If different, how and why justified?

Reply: Thank you for your comments. ALCC in our manuscript means 'Anthropogenic Land-Cover Change'. We have added the definitions of ALCC in our revised manuscript (**Page 3 line 15**).

Page 3 line 8: PAGELandCover6k mostly focuses on paleoclimate indicators (e.g. pollen) and not exclusively on regional patterns. Here you focus on small region (Scandinavia) with unusually-good historical records? How does this work fit with PAGES paleoclimate projects?

Reply: Thank you for your comments. One of the LandCover6k's aims is to "evaluate the existing ALCC (anthropogenic land-cover change) scenarios with the combined information from the pollen-based reconstructions, archeological and historical data, and other evidence of human-induced land-cover change such as paleofire reconstructions" (Gaillard et al., 2015). We aim to evaluate the existing ALCC scenarios in Scandinavia based on historical records. Thus, this work fits with LandCover6k projects.

Reference: Gaillard, M. J., and LandCover6k Interim Steering Group members: LandCover6k: Global anthropogenic land-cover change and its role in past climate, Past Global Change Magazine, 23, 38-39, 2015.

Page 3 line 10: "Errors" in regional reconstructions or in global products. Need clarity here.

Reply: Thank you for your comments. We have changed "Errors" to "Errors in global products" (**Page 3 line 20**).

Page 3 line 16: farmers are were, please make careful and consistent use of past tense.

Reply: Thank you for your comments. We have changed "are" to "were". Please see **page 3 line 27**.

Page 3 line 30 to page 4 line 1: "importance …. could fail to be determined precisely" What? Confusing!

Reply: Thank you for your comments. We have changed the sentence to "The impacts of ALCC on climate and environment cannot be evaluated precisely without grid-based cropland maps". Please see **page 4 lines 11-12**.

Page 4 lines 15 to 19 - finally, a clear statement of intent. This text could replace much of what precedes it. Dataset will provide? Better: dataset provides!

Reply: Thank you for your comments. We have changed "will provide" to "provides" (Please see **page 5 line 1**).

Overall, good helpful description but methods, data and results sections need careful scrutiny and occasional re-write!

Page 36, around line 30: Reference list not in alphabetical order. Please check entire reference list for similar errors.

Reply: Thank you for your comments. We have checked entire references list and the references list is in alphabetical order now.

Typesetters and proofreaders from Copernicus will apply very careful very good language services for this manuscript but they will have many questions! Two changes suggested here: careful reading

and re-writing by a native English speaker and careful definition of all acronyms (consider a list of acronyms as suggested) will make their job easier and your product better!

Reply: Thank you for your valuable comments for our manuscript. We have defined all acronyms (Please see **Appendix**). The manuscript has been read and polished by a native English speaker. But we believe language services from Copernicus will make our product better.